# UTILITY-BASED ADAPTIVE TEACHING STRATEGIES USING BAYESIAN THEORY OF MIND

## ABSTRACT

Good teachers always tailor their explanations to the learners. Cognitive scientists model this process under the rationality principle: teachers try to maximise the learner's utility while minimising teaching costs. To this end, human teachers seem to build mental models of the learner's internal state, a capacity known as Theory of Mind (ToM). Inspired by cognitive science, we build on Bayesian ToM mechanisms to design teacher agents that, like humans, tailor their teaching strategies to the learners. Our ToM-equipped teachers construct models of learners' internal states from observations and leverage them to select demonstrations that maximise the learners' rewards while minimising teaching costs. Our experiments in simulated environments demonstrate that learners taught this way are more efficient than those taught in a learner-agnostic way. This effect gets stronger when the teacher's model of the learner better aligns with the actual learner's state, either using a more accurate prior or after accumulating observations of the learner's behaviour. This work is a first step towards social machines that teach us and each other, see `https://teacher-with-tom.github.io`.

## 1 INTRODUCTION

When tasked with imparting an understanding of the solar system, a physics teacher tailors their explanation based on the audience. The approach taken for a 10-year-old astrophysics enthusiast differs significantly from that employed for an advanced master's student. In fact, the teacher provides an explanation that maximises the likelihood of the listener understanding the concept. This pedagogical sampling phenomenon has been explored in cognitive science notably in Gweon et al. (2018). This study involves children being asked to demonstrate the use of a toy to knowledgeable or ignorant children learners. It shows that the behaviour of the teacher-child depends on prior observations of the learner-child. Specifically, if the learner has previously interacted with a similar toy in the presence of the teacher, the teacher only exhibits partial functionality of the toy. Conversely, when no prior interaction is observed, the teacher demonstrates the complete use of the toy.

By definition, the aim of a teacher is to ensure the learner's understanding. An option for the teacher would be to demonstrate the full functionality of the toy each time, but this comes with a cost. Rather, the teacher strikes a balance between the learner's understanding, reflected in its subsequent behaviour, and the costs of teaching. Assuming the teacher is rational, we can thus consider that this trade-off is the teacher's *utility* (Goodman & Frank, 2016; Jara-Ettinger et al., 2016). Importantly, learners also evaluate the teacher based on its actions (Bass et al., 2022) teachers who solely provide the missing information for the learner to achieve the task are also perceived as more trustworthy than over-informative ones (Gweon et al., 2018).

More generally, human teachers choose how to teach based on a prediction of how their guidance signal will be received, as outlined in the Inferential Social Learning (ISL) framework (Gweon, 2021). In this framework, humans acquire knowledge by making inferences from observing others' behaviour and leverage this knowledge to help others learn. More precisely, ISL is grounded on a set of cognitive mechanisms constituting the Theory of Mind (ToM), which refers to the human ability to understand and predict the actions of others by inferring their mental states, such as prior knowledge, goals, intentions, beliefs etc. (Baker & Saxe, 2011). ToM can be understood as the inverse planning of an intuitive behavioural model predicting what others would do given their mental state (Baker et al., 2009). To be efficient, human pedagogical interventions such as selection of

examples (Shafto et al., 2014) or demonstrations (Ho et al., 2021) require ToM. ISL is considered a key component to humans mutual understanding as well as a foundation of humans' powerful capacity to efficiently learn from others. Therefore, incorporating ISL mechanisms into AI systems is a promising way to make human–machine interactions more informative, productive, and beneficial to humans (Gweon et al., 2023; Sigaud et al., 2022).

In this paper, we introduce teacher agents equipped with a ToM model of the learner agent's internal state, including its goal, intention, belief, and sensory capacity. The goal of this work is to study whether learner-specific teachers who model the learner's internal state are more efficient than learner-agnostic ones and more importantly to explore the limitations of ToM models with inaccurate priors or limited observation of the learner, in a context where providing guidance incurs a cost proportional to its informativeness.

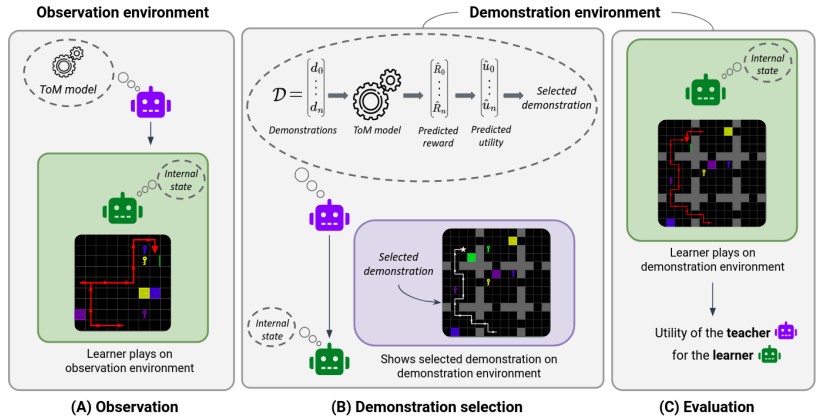

Figure 1: (A) The teacher observes a learner with a particular internal state behaving in a simple environment $\mathcal{M}^{\mathrm{obs}}$ and infers a ToM model of this learner. (B) In a more complex environment $\mathcal{M}^{\mathrm{demo}}$, the teacher uses this ToM model to predict the usefulness for the observed learner of each demonstration of a provided dataset $\mathcal{D}$, out of which it selects the utility-optimal demonstration $d^*$. The learner observes $d^*$ and updates its knowledge about $\mathcal{M}^{\mathrm{demo}}$. (C) The learner behaves in $\mathcal{M}^{\mathrm{demo}}$ and receives a reward. The teacher is evaluated on the utility of $d^*$, which is the learner's reward minus the cost incurred by the teacher in delivering that demonstration.

To achieve this, as depicted in Figure 1, we define *ToM-teachers* able to

1. update a *belief* about the internal state (i.e. goal, intention, belief, sensory capacity) of an unknown learner through Bayesian inference based on observations of its behaviour in a simple environment, see Figure 1(A), and
2. leverage this belief to estimate the utility of different demonstrations in a more complex environment, similarly to human planning as described in Ho et al. (2022), in order to select the most effective one for the specific observed learner, see Figure 1(B).

To conduct our experiments, we present two environments: a toy environment reminiscent of Gweon's study mentioned above (Gweon et al., 2018), and a more challenging gridworld environment for goal-conditioned 2D navigation, see Figure 1. Depending on its sensory capacity, the learner might require the help of a teacher agent providing a demonstration showing the locations of the objects needed to complete the task. However, the teacher ignores the goal of the learner and its sensory capacity, but can infer them from a past trajectory of the learner in a simpler environment.

In this setup, the teacher must select the most useful demonstration providing enough information to help the learner reach its goal, but at a minimal teaching cost. The demonstration utility is optimal if it contains the necessary and sufficient amount of information for the learner to reach its goal. In this context, we show that the teacher must display accurate ISL abilities, inferring the learner's goal and sensory capacity from the past trajectory to effectively assist the learner. While this result might not be surprising, we further find, on the other hand, that some learner-agnostic teaching strategies outperform ToM-teachers when inaccurate prior of the learner's policy and/or limited observation of its behaviour are available.

## 2 RELATED WORK

In addition to cognitive science researches on human pedagogy (Shafto et al., 2014; Gweon, 2021; Ho et al., 2021), this work is related to the following interconnected research areas:

**Theory of Mind (ToM):** Observer agents capable of inferring the internal state, including the goal, of another agent have been developed based on Bayesian Inference (Ying et al., 2023; Reddy et al., 2018) and neural networks (Rabinowitz et al., 2018; Nguyen et al., 2022). The introduction of a ToM model of the teacher used by the learner to modulate guidance has demonstrated benefits in the learning process, as shown in Peltola et al. (2019). However, these works do not explore how to leverage these models of ToM for the teacher to assist the learner in achieving its goal, as human teachers do, as explained in Ho et al. (2022).

**Machine teaching:** Machine Teaching is formalised as the problem of identifying the minimal teaching signal maximising the learner's reward (Zhu et al., 2018; Brown & Niekum, 2019). The teacher possesses knowledge of the learner's goal and aims to either generate the teaching data (Zhu, 2013) or to extract it from a dataset (Yang & Shafto, 2017), helping the learner agent achieve its goal. A teaching signal is considered optimally useful if it maximises utility, that is it enables the learner to achieve its goal while minimising the teaching cost (Zhu et al., 2018). In our framework the teacher must select the most helpful demonstration from a given set for various types of learner. Yet, unlike these prior studies, our teacher assists various learners with different goals and sensory capacities, and thus different optimal demonstrations. Previous studies have demonstrated the benefits of adaptivity in sequential machine teaching (Chen et al., 2018) and motor control (Srivastava et al., 2022) for learning. Unlike this prior research, we introduce a model of ToM explicitly modeling the learner's mental state as a pivotal component of our teacher's adaptivity. The demonstration selection strategy of our teacher is similar to the one used in cognitive science to model human's strategy as described in Ho et al. (2022): it uses the learner's ToM model to predict the outcomes of different possible demonstrations for the learner, in order to select the demonstration of optimal utility.

**Bayesian Inference:** Bayesian Inference is a widely used mechanism for inferring the goals of other agents by computing posterior probabilities based on their actions and policies (Baker et al., 2009; Baker & Saxe, 2011; Zhi-Xuan et al., 2020; Ying et al., 2023). In our work, we employ it as a tool to infer the internal state of the learner, including its goal and sensory capacity. In Shafto et al. (2012); Bass et al. (2022), Bayesian ToM models were conversely used by the learner to infer the internal state of the teacher. Additionally, similarly to Zhu (2013); Ho et al. (2022), we assume a Bayesian learner to ensure direct communication from the teacher to the learner as the demonstration selected by the teacher modifies the belief of the learner about the environment.

## 3 METHODS

Our general framework is depicted in Figure 1. Below we describe the components in more details.

### 3.1 LEARNING ENVIRONMENT

We introduce the learners' environment as a Goal-Conditioned Partially Observable Markov Decision Process (GC-POMDP), which is a combination of a Goal-Conditioned Markov Decision Process (GC-MDP) and, similarly to Rabinowitz et al. (2018), a Partially Observable Markov Decision Process (POMDP). In GC-POMDPs, agents aim at achieving different goals with limited information on the current state of the environment. An instance $\mathcal{M}^j$ of a GC-POMDP is defined by:

- A set of states $\mathcal{S}^j$, a set of possible actions $\mathcal{A}^j$, a transition function $\mathcal{T}^j : \mathcal{S}^j \times \mathcal{A}^j \to \mathcal{S}^j$,

- A set of possible goals $\mathcal{G}^j$,

- A history-dependent goal-conditioned reward function $R^j : \mathcal{H}^j \times \mathcal{G}^j \to \mathbb{R}$, where $\mathcal{H}^j$ is the space of histories. We define a *history* as a sequence of state-action pairs over time, which can be formulated as $\mathcal{H}^j = \bigcup_t \mathcal{H}^j_t$ in which $\mathcal{H}^j_t = \{(s_0, a_0, \ldots, s_{t-1}, a_{t-1})\} = \prod_t (S^j \times \mathcal{A}^j)$.

We consider that all GC-POMDPs share their action and goal spaces denoted $\mathcal{A}$ and $\mathcal{G}$. In summary, a GC-POMDP is defined as $\mathcal{M}^j = (\mathcal{S}^j, \mathcal{A}, \mathcal{T}^j, \mathcal{G}, R^j)$.

In practice, our GC-POMDPs are different instances of similar gridworld environments constructed from the `MiniGrid` library (Chevalier-Boisvert et al., 2023). Another example with a toy environment is described in Appendix A.

## 3.2 LEARNER

We consider a finite family of agents $\mathcal{L} = \{L_i, i \in I\}$ that we call *learners*. A learner $L_i$ is defined by a goal $g_i \in \mathcal{G}$ and an observation function $v_i$, i.e. $L_i = (g_i, v_i)$.

In an environment $\mathcal{M}^j = (\mathcal{S}^j, \mathcal{A}, \mathcal{T}^j, \mathcal{G}, R^j)$, the observation function is defined on the state space towards an observation space $\Omega_i$, $v_i : \mathcal{S}^j \to \Omega_i$. The set of observation functions is denoted $\mathcal{V}$ and is assumed to be identical for all the considered GC-POMDPs. The aim of the learner is to maximise the reward functions $R^j$, conditioned on the learner's goal $g_i$. In practice, the learner must achieve its goal in minimum time to maximise its reward. We characterise the behaviour of a learner $L_i$ on $\mathcal{M}^j$ as a trajectory $\tau_i = \{(s_t, a_t^i) \in \mathcal{S}^j \times \mathcal{A}\}_{t=0}^T$. For the same trajectory, two learners $L_i$ and $L_{i'}$ with different observation functions $v_i \neq v_{i'}$ acquire different knowledge about the environment, and two learners with different goals $g_i \neq g_{i'}$ receive different rewards.

In POMDPs, since the state is not directly observed, the learner must rely on the recent history of observations, to infer a distribution over states and maintain a belief on the environment state (Kaelbling et al., 1998; Ghavamzadeh et al., 2015). To model learner's $L_i$ policy, we thus consider at every step $t$ its *belief* $b_t^{i,j}$ over a set of possible states $\mathcal{S}_B^j$ of environment $\mathcal{M}^j$. We assume that the support of the belief contains the real state space, $\mathcal{S}^j \subset \mathcal{S}_B^j$ and note $\mathcal{B}^j$ the continuous space of beliefs.

At every step $t$, the environment being in a state $s_t \in \mathcal{S}^j$ and the observation being $o_t^i = v_i(s_t)$, the belief of learner $L_i$ about the state $s \in \mathcal{S}_B^j$ of the environment is updated using Bayesian update:

$$\forall s \in \mathcal{S}_B^j, \quad b_{t+1}^{i,j}(s) = \frac{b_t^{i,j}(s) \times \mathbb{P}(o_t^i|s)}{\int_{s' \in \mathcal{S}_B^j} b_t^{i,j}(s') \times \mathbb{P}(o_t^i|s')}. \tag{1}$$

Unless mentioned otherwise, we assume that the learner's initial belief $b_0^{i,j}$ on the state of $\mathcal{M}^j$ is uniform over the set of possible states $\mathcal{S}_B^j$. In the experiments presented below, we additionally assume that all learners share a policy on the environment $\mathcal{M}^j$ conditioned by a goal, an observation function and a belief:

$$\pi^j(.|g, v, b^L) : \cup_i \Omega_i \times \mathcal{A} \to [0, 1], \quad \text{with } (g, v, b^L) \in \mathcal{G} \times \mathcal{V} \times \mathcal{B}^j. \tag{2}$$

To simulate a trajectory $\tau^i$ of learner $L_i$ on $\mathcal{M}^j$, one only needs to know the tuple $(\pi^j, g_i, v_i, b_0^{i,j})$. In practice, the learners use a single policy denoted $\pi$ for all the considered GC-POMDPs.

Moreover, within MiniGrid environments, the observation functions $v_i$ are defined by a square area of size $v_i \times v_i$ cells, known as the *receptive field* of learner $L_i$. This receptive field defines the localised region in front of the learner, mimicking visual sensory capacities and a larger receptive field size helps the learner reach its goal faster.

## 3.3 TEACHER

We introduce an agent called *teacher* whose aim is to optimally help the learner maximise its reward on a GC-POMDP $\mathcal{M}^{\text{demo}} = (\mathcal{S}^{\text{demo}}, \mathcal{A}, \mathcal{T}^{\text{demo}}, \mathcal{G}, R^{\text{demo}})$ by providing a demonstration.

### 3.3.1 UTILITY BASED DEMONSTRATION SELECTION STRATEGY

We define a demonstration of length $n \in \mathbb{N}$ on $\mathcal{M}^{\text{demo}}$ as a sequence of actions $d = (a_0^{\text{demo}}, \dots, a_{n-1}^{\text{demo}}) \in (\mathcal{A})^n$. We consider the demonstration to be provided as if the teacher were *teleoperating* the learner as described in Silva & Costa (2019). Thus, at step $t$ of the demonstration, learner $L_i$ observes $\bar{o}_{t+1}^i = v_i \left( \mathcal{T}_{\text{demo}}(s_t, a_t^{\text{demo}}) \right)$. Following the same demonstration leads to varying observation sequences for learners with different observation functions. The learner's belief about the new environment $\mathcal{M}^{\text{demo}}$ is updated based on the observations $(\bar{o}_1^i, \dots, \bar{o}_n^i)$ resulting from the demonstration, as in Equation 1 and depicted in Figure 1(B).

This updated belief is then used as initial belief $b_0^{i,\text{demo}}$ by the learner. In other words, the aim of the demonstration is to provide to the learner a prior knowledge about the new environment. The environment is then reset to its initial state, and the learner behaves following a policy $\pi^{\text{demo}}$ defined in Equation 2 starting with belief $b_0^{i,\text{demo}}$. As shown in Figure 1(C), the execution of this policy produces a trajectory $\tau^{\text{demo}} = \{(s_t^{\text{demo}}, a_t^{\text{demo}})\}_{t=0}^{T}$ where $T \in \mathbb{N}$ and the learner receives a reward $R^{\text{demo}}(\tau^{\text{demo}}, g_i)$ denoted $R^{\text{demo}}(L_i|d)$, which represents the reward of learner $L_i$ on environment $\mathcal{M}^{\text{demo}}$ after having observed demonstration $d$.

We assume that the teacher knows the environment $\mathcal{M}^{\text{demo}}$ and has access to a set of potential demonstrations $\mathcal{D}$ to be shown on $\mathcal{M}^{\text{demo}}$ as well as a teaching cost function $c_\alpha : \mathcal{D} \to \mathbb{R}$ parameterised $\alpha \in \mathbb{R}_+$. For a given parameter $\alpha$, the cost of a demonstration $d \in \mathcal{D}$, denoted $c_\alpha(d)$, represents the cost for the teacher of showing demonstration $d$ to a learner. In our context, this function increases with the length of the demonstration.

We introduce on the environment $\mathcal{M}^{\text{demo}}$ the *utility* of a demonstration $d$ for a learner $L_i$ as the reward of the learner after having observed the demonstration $d$ on $\mathcal{M}^{\text{demo}}$ minus the cost for the teacher of showing this demonstration: $u_\alpha^{\text{demo}}(d, L_i) = R^{\text{demo}}(L_i|d) - c_\alpha(d)$. The aim of the teacher is to select the demonstration $d_i^*$ that maximises the utility for the learner $L_i$:

$$d_i^* = \arg\max_{d \in \mathcal{D}} \quad \underbrace{u_\alpha^{\text{demo}}(d, L_i)}_{R^{\text{demo}}(L_i|d) - c_\alpha(d)} \quad . \tag{3}$$

However, the teacher does not know neither the learner's goal $g_i$ nor its observation function $v_i$. Instead, it can only access a past trajectory $\tau^{\text{obs}}$ of the same learner $L_i$, but in a different environment $\mathcal{M}^{\text{obs}} = (\mathcal{S}^{\text{obs}}, \mathcal{A}, \mathcal{T}^{\text{obs}}, \mathcal{G}, R^{\text{obs}})$, see Figure 1(A). Therefore, in order to approximate Equation 3, the teacher should estimate the utility of each demonstration $d$ in $\mathcal{D}$ for this learner, see Figure 1(B). As the teacher knows the teaching cost function, this is equivalent to estimating the learner's reward.

### 3.3.2 TEACHING ENVIRONMENT

Teaching an unknown learner $L_i = (g_i, v_i)$ can be formalised as maximising a reward function in a POMDP framework (Rafferty et al., 2015; Yu et al., 2023) which can be simplified in the case of demonstration selection into a contextual Multi-Arms bandit (MAB) (Clément et al., 2015). Our approach involves a teaching MAB relying on a pair of environments $(\mathcal{M}^{\text{obs}}, \mathcal{M}^{\text{demo}})$. The teaching state space is the set of all possible learners $\mathcal{L} = \mathcal{G} \times \mathcal{V}$. The MAB being in state $L_i$, the observation function $\mathcal{O}^{\text{obs}}$ generates a context ($\tau^{\text{obs}} = \{(s_k, a_k^{\text{obs}})\}_{k=0}^{K-1}, b_0^{L_i}) \in \Delta^{\text{obs}}$ which corresponds respectively to a trajectory of learner $L_i$ within the environment $\mathcal{M}^{\text{obs}}$ and the learner's initial belief. The teaching action space is the available set of demonstrations $\mathcal{D}$ on $\mathcal{M}^{\text{demo}}$. The reward function is the utility $u_\alpha^{\text{demo}}$ defined on the environment $\mathcal{M}^{\text{demo}}$ which takes as arguments a state (the learner's internal state) and an action (a demonstration). The teaching contextual MAB is therefore defined as $\mathcal{E} = \{\mathcal{L}, \mathcal{D}, \mathcal{O}^{\text{obs}}, \Delta^{\text{obs}}, u_\alpha^{\text{demo}}\}$.

### 3.3.3 BAYESIAN ToM-TEACHER

To estimate the utility $u_\alpha^{\text{demo}}(d, L_i)$ of a demonstration $d$ in the teaching MAB $\mathcal{E}$ in state $L_i$, we introduce a teacher equipped with a ToM model that we refer to as *ToM-teacher*. In our case, the ToM is used to model the MAB state (learner's hidden internal state) from an observation (past trajectory and initial belief), leading to the estimation of the teaching MAB reward function that is the utility function over the set of demonstrations for the unknown learner $L_i$.

We present a ToM-teacher using Bayesian inference, called *Bayesian ToM-teacher*. We assume that the teacher has access to a behavioural model of the learners – that is an approximation of their policy $\hat{\pi}$ – along with a support for the teaching MAB state constituted by sets of possible goals $\mathcal{G}_B$ and observation functions $\mathcal{V}_B$. We make the assumption that these spaces are discrete and that both sets contain the real sets of goals and observation functions ($\mathcal{G} \subset \mathcal{G}_B$ and $\mathcal{V} \subset \mathcal{V}_B$).

From an observation of the teaching MAB state, $\mathcal{O}^{\text{obs}}(L_i) = (\tau^{\text{obs}}, b_0^{L_i})$, the Bayesian ToM-teacher computes a belief $b^T$ about the teaching MAB state, that is a probability distribution over the joint space $\mathcal{G}_B \times \mathcal{V}_B$. At step $k \in [0, K-1]$ of the observed trajectory $\tau^{\text{obs}}$, for every pair $(g, v) \in \mathcal{G}_B \times \mathcal{V}_B$, it derives from Equation 1 and the observed initial belief $b_0^{L_i}$, the belief that a learner

would have with observation function $v$ after producing the trajectory $\tau^{\text{obs}}[0:k-1]$, denoted $b_k^{v,\text{obs}}$. It then updates its own belief about the learner goal and observation function based on the Bayesian update rule:

$$\forall (g,v) \in \mathcal{G}_B \times \mathcal{V}_B, \quad b_{k+1}^T(g,v) = \frac{b_k^T(g,v) \times \hat{\pi}\left(v(s_{k-1}), a_k^{\text{obs}}|g, b_k^{v,\text{obs}}\right)}{\sum_{g' \times v' \in \mathcal{G}_B \times \mathcal{V}_B} b_k^T(g',v') \times \hat{\pi}\left(v'(s_{k-1}), a_k^{\text{obs}}|g', b_k^{v',\text{obs}}\right)}. \tag{4}$$

The quantity $b_k^T(g,v)$ represents the probability of the learner having a goal $g$ and an observation function $v$, given that it produced trajectory $\tau^{\text{obs}}[0:k-1]$, under the assumption that, to generate $\tau^{\text{obs}}[0:k-1]$, the learner follows policy $\hat{\pi}$. The final belief $b_K^T(g,v)$ represents the probability that the teaching MAB is in state $L = (g,v)$.

The teacher estimates the utility of a demonstration $d \in \mathcal{D}$ in the teaching MAB $\mathcal{E}$ in state $L_i$ by computing the expected value:

$$\hat{u}_\alpha^{\text{demo}}(d) = \sum_{(g,v) \in \mathcal{G}_B \times \mathcal{V}_B} \hat{u}_\alpha^{\text{demo}}(d, L = (g,v)) \times b_K^T(g,v), \tag{5}$$

where $\hat{u}_\alpha^{\text{demo}}(d, L)$ is the estimated utility of demonstration $d$ for a teaching MAB in state $L$. To compute this quantity, the teacher computes the belief $b_0^{v,\text{demo}}$ of a learner $L = (g,v)$ on $\mathcal{M}^{\text{demo}}$ after having observed demonstration $d$, based on Equation 1 and the observed initial belief $b_0^{L_i}$. From the tuple $(\hat{\pi}, g, v, b_0^{v,\text{demo}})$, the teacher simulates a trajectory $\hat{\tau}^{\text{demo}}$ and computes the associated estimated reward $\hat{R}^{\text{demo}}(L|d) = R^{\text{demo}}(\hat{\tau}^{\text{demo}}, g)$ leading to the estimated utility $\hat{u}_\alpha^{\text{demo}}(d, L) = \hat{R}^{\text{demo}}(L|d) - c_\alpha(d)$. The expected utility can be expressed as the expected reward of the unknown learner after following demonstration $d$ minus the cost of the demonstration:

$$\hat{u}_\alpha^{\text{demo}}(d) = \underbrace{\left( \sum_{(g,v) \in \mathcal{G}_B \times \mathcal{V}_B} \hat{R}^{\text{demo}}(L = (g,v)|d) \times b_K^T(g,v) \right)}_{\text{Expected reward}} - c_\alpha(d). \tag{6}$$

The teacher selects the greedy demonstration $d^*$ over the estimated utility of the teaching MAB $\mathcal{E}$ in state $L_i$, approximating Equation 3 with $d^* = \arg\max_{d \in \mathcal{D}} \hat{u}_\alpha(d)$.

We define two ToM-teachers which differ in their prior model of the learner's policy $\hat{\pi}$:

• The *aligned ToM-teacher* possesses exact knowledge of the learner's policy, $\hat{\pi} = \pi$.

• The *rational ToM-teacher (with parameter $\lambda$)* only assumes that the learner is rational, meaning it tries to reach the goal in minimum time, but its approximate policy $\hat{\pi} \neq \pi$ is based on a Boltzmann policy that considers the expected distance between the learner and the goal after taking different actions. The temperature parameter $\lambda$ of the Boltzmann policy represents the assumed degree of rationality of the learner in terms of how much the learner favours actions towards its goal, see Appendix B.3 for more details.

## 4 EXPERIMENTS

**Environments:** The observation environment $\mathcal{M}^{\text{obs}}$ is a $11 \times 11$ MiniGrid gridworld (Chevalier-Boisvert et al., 2023) and is enclosed by walls along its borders. The environments contains four door-key pairs of colours in the set $\mathcal{G} = \{green, blue, purple, yellow\}$. To open a door, an agent has to possess the key of the same colour. We study the influence of the observation environment's size on the accuracy of the ToM models in Appendix G.

The demonstration environment $\mathcal{M}^{\text{demo}}$, contains the same objects but over $33 \times 33$ cells. It is composed of nine rooms of $11 \times 11$ cells, separated by walls. In both environments, a trajectory stops either when the learner opens its goal door or when the maximum number of actions is elapsed.

**Learner:** The learner's goal is to open a door as fast as possible. We use the default goal-conditioned trajectory reward function of the MiniGrid environments: $R(\tau, g) = 1 - 0.9 \times \frac{\text{length}(\tau)}{\text{max\_steps}}$ if the door

of colour $g \in \mathcal{G}$ is open at the end of trajectory $\tau$, and $R(\tau, g) = 0$ otherwise. In $\mathcal{M}^{\text{obs}}$, we set max_steps $= 11^2 = 121$, and in $\mathcal{M}^{\text{demo}}$, we use max_steps $= \frac{33^2}{2} = 544$.

The learner possesses either a view with dimensions $v \times v$ cells with $v \in \{3, 5\}$ or full observability ($v = full\_obs$) of the environment. With $v \neq full\_obs$, the learner does not see behind the walls.

We define the learner's policy as a decision tree (Appendix B.1). We assume that the learner attempts to reach the key before trying to open the door and acts greedily when it knows the location of the objects and actively explores otherwise. The greedy policy follows the shortest path computed by the $A^*$ algorithm (Hart et al., 1968) within the known parts of the environment. The active exploration policy selects actions best reducing the uncertainty on the environment state.

**Teachers:** As defined above in Section 3.3, we consider two teachers equipped with a ToM model of the learner, an ■ *aligned ToM-teacher* and a ■ *rational ToM-teacher* with a parameter $\lambda$. We compare the utilities of their demonstrations to that of 5 baseline teachers, one for upper-bound and four learner-agnostic teachers which do not leverage the past observations of the learner in their strategies for demonstration selection:

■ The *omniscient teacher* knows the actual goal, observation function and policy of the learner and provides the utility-optimal demonstration. It sets an upper-bound teacher's utilities.

■ The *reward-optimal non-adaptive teacher* selects the demonstration in $\mathcal{D}$ maximising the mean reward over all the possible learners without considering the teaching cost. In practice, this teacher provides the demonstration showing all the objects (keys and doors) of the environment.

■ The *utility-optimal non-adaptive teacher* selects the demonstration in $\mathcal{D}$ maximising the mean utility over all possible learners.

■ The *uniform modelling teacher* uniformly samples a learner in $(g, v) \in \mathcal{L}$ and provides the demonstration maximising the utility for $L = (g, v)$.

■ The *uniform sampling teacher* selects a demonstration uniformly among the set $\mathcal{D}$ of available demonstrations. This teacher does not have any model of the learner.

**Demonstration set:** The demonstration set $\mathcal{D}$ contains shortest demonstrations for each pairs $(g, v) \in \mathcal{G} \times \mathcal{V}$ showing the learner's key and door goal at a distance of at least $v$. In addition, we generate demonstrations showing $N \in [3, 8]$ random objects (key or door) of the environment, see Appendix B.2 for details. We use a linear teaching cost with parameter $\alpha = 0.6$ normalised by the size $l_{max}$ of the longest demonstration of $\mathcal{D}$. For a demonstration of length $l_d$, the teaching cost is $c_\alpha(l_d) = \alpha \times \frac{l_d}{l_{max}}$. In practice, the longest demonstration is the one showing all objects, $N = 8$.

**Metrics:** A teacher is evaluated based on the measured utility of the demonstration it has selected for the observed learner $L$, given by $u_\alpha^{\text{demo}}(d^*, L) = R^{\text{demo}}(L|d^*) - c_\alpha(d^*)$.

**Experiments:** We conducted 100 experiments for each pair $(g, v) \in \mathcal{G} \times \mathcal{V}$. Mean utilities of demonstrations selected by teachers for learners with a fixed receptive field size $v$ are in Figure 2 and Appendix CTable1. Computed over 400 trials with a 95% confidence interval, Student T-tests assess significant differences between mean utilities of two teachers. Environments, both observation and demonstration, are randomly generated in each trial. All teachers operate within the same environment pair ($\mathcal{M}^{\text{obs}}, \mathcal{M}^{\text{demo}}$), selecting demonstrations from the same set $\mathcal{D}$, while ToM-teachers observe the same learner trajectory on $\mathcal{M}^{\text{obs}}$.

## 5 RESULTS

We provide results when the learners are observed under two conditions: for a full episode or for only their 10 first actions, leading to more uncertain inference about their goals and sensory capacities.

### 5.1 OBSERVING A FULL TRAJECTORY OF THE LEARNER

Figure 2 illustrates the mean utility of the demonstrations selected by each teacher, for learners with varying receptive field sizes acting in $\mathcal{M}^{\text{obs}}$ during a full episode.

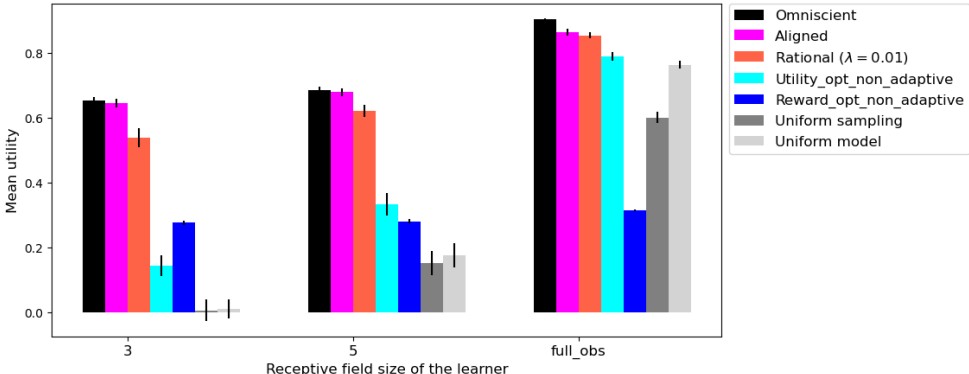

Figure 2: Mean utilities and 95% confidence interval of ToM-teachers (rational teacher with parameter $\lambda = 0.01$) and baseline teachers for learners with varying receptive field sizes of $[3, 5, full\_obs]$ observed on $\mathcal{M}^{\text{obs}}$ during a full episode.

Across all the considered learners with varying receptive field sizes, the demonstrations chosen by the ToM-teachers outperform those of learner-agnostic baseline teachers. As the task difficulty increases for the learner (i.e., when its receptive field size decreases), the learner requires both more informative and more specific demonstrations to achieve its goal. Consequently, having an accurate model of the learner becomes necessary to ensure the selection of helpful demonstrations.

The mean utility of aligned ToM-teachers is not significantly different from that of the omniscient demonstrations (p-values $> 0.3$)[1] for learners with receptive field of sizes 3 and 5. In contrast, uniform teachers select demonstrations with close-to-null mean utility for learners with a receptive field size of 3 and demonstrations that are four times less useful than those of the ToM-teachers for learners with receptive field size of 5. The utility-optimal and reward-optimal non-adaptive teachers perform at most half as well as the ToM-teachers for these learners, see Appendix C Table 1.

On the contrary, as the task becomes easier for the learners (with wider sensory capacities), the mean utilities of the demonstrations selected by learner-agnostic teachers get closer to those of the ToM and omniscient teachers' demonstrations, as the need for selecting a specific demonstration based on an accurate model of the learner decreases. In fact, with full observability, any demonstration from the demonstration set suffices for the learner to reach the goal.

With a teaching cost of $\alpha = 0.6$ it is worth noting that the utility-optimal non-adaptive teacher tends to select less informative demonstrations (with low teaching cost) leading to higher mean utility for learners with full observability and lower mean utility for learners with a limited view. Selecting the demonstration maximising the mean reward over the learners proves to be too expensive and consistently results in poor utility. We further discuss the teaching cost parameter in Appendix F.

The precision of the ToM-teacher's behavioural model of the learner (i.e. its policy) directly impacts the utility of the selected demonstrations. The aligned ToM-teacher selects more beneficial demonstrations on average than the rational ToM-teacher which relies on an approximation of the learner's policy, for learners with receptive field of sizes 3 and 5 (p-values $< 0.01$) and their utilities are not significantly different for learner with full observability (p-value $> 0.15$), see Appendix C Table 1.

A high degree of accuracy of the ToM-teacher's model of the learner's behavioural policy enhances belief updates of Equation 4, resulting in more accurate modelling of the learner's internal state. To illustrate this, we derive in Appendix D explicit inferences regarding the learner's goal and receptive field size from ToM-teachers beliefs featuring varying degrees of accuracy.

## 5.2 LIMITED OBSERVATION OF THE LEARNER

Now, instead of having access to the entire trajectory $\tau^{\text{obs}}$ of the learner in $\mathcal{M}^{\text{obs}}$, the teacher only has access to its first 10 actions, that is the partial trajectory $\tau^{\text{obs}}[: 10]$.

---

[1] A t-test with null hypothesis $H_0$: there is no significant difference between the utilities of both teachers.

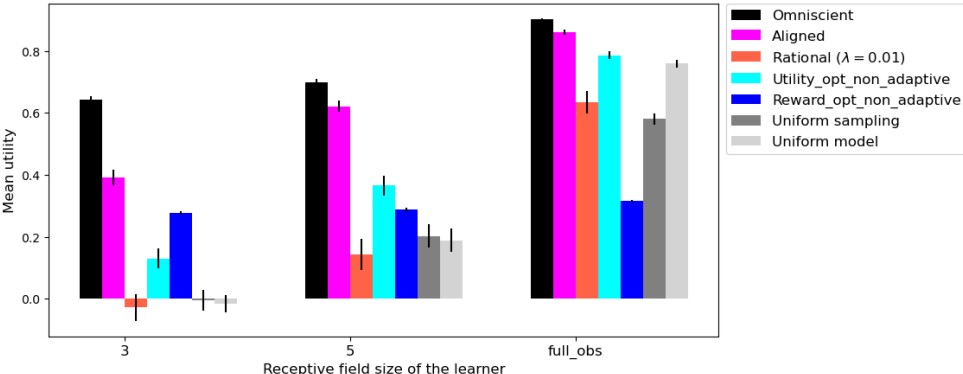

Figure 3: Mean utilities and 95% confidence interval of teachers as in Figure 2 observed on $\mathcal{M}^{\text{obs}}$ during the 10 first steps of an episode ($\tau^{\text{obs}}[:10]$).

As expected, with limited information about the learner, both ToM-teachers select demonstrations achieving mean utilities that are further away from the utility of the omniscient teacher's demonstrations. Nonetheless, the aligned ToM-teacher still outperforms the learner-agnostic teachers on average for all the considered learners, as depicted in Figure 3.

However, relying solely on the hypothesis that the learner is highly rational is not enough to accurately model its internal state when having access to limited observation of its behaviour. In fact, the utility of the demonstration selected by the rational ToM-teacher with low temperature parameter $\lambda = 0.01$ decreases approximately by $100\%$, $75\%$ and $25\%$ for learners with receptive field sizes of 3, 5 and full observability, see Appendix C Table 2. As detailed in Appendix F E, with the approximate learner's policy, the rational ToM-teacher misinterprets the learner's behaviour. This leads to incorrect conclusions about the learner's internal state and, consequently, inaccurate demonstration selection. As a result, the performance of the rational teacher is not significantly different from that of the uniform modelling teacher for learners with limited view (p-values $> 0.15$) but significantly lower for learners with full observability (p-value $< 0.01$).

Furthermore, in this limited information context, providing the demonstration maximising the mean utility on all the learners proves to be more useful that relying on an imprecise behavioural model of the learner. For all considered learners, the utility-optimal non-adaptive teacher significantly outperforms the rational ToM-teacher (p-values $< 0.01$), see Appendix C Table 2.

## 6 CONCLUSION AND FUTURE WORKS

In this work, we have studied the integration of ISL mechanism for teaching learners with different goals, beliefs or sensory capacities. We integrated a Theory of Mind model using Bayesian inference into a teacher agent to infer the learner's internal state and adapt its teaching strategy. We demonstrated that leveraging this ToM model, combined with a behavioural model of the learner, is more efficient than adopting learner-agnostic teaching strategies. We also explored the limitations of ToM models with limited observation of the learner and approximate behavioural models. In summary, we have shown that machine ISL can enhance knowledge transmission between AI systems, and we are convinced that it represents a pathway toward richer and more trustworthy knowledge exchange between AI systems and humans (Gweon et al., 2023; Sigaud et al., 2022).

There are many exciting directions for future work, particularly towards more tractable models of ToM mechanisms in higher-dimensional environments, for example, using variational methods (Zintgraf et al., 2020) or ensembling to approximate Bayesian inference. Another direction for future research is to employ reinforcement learning to train the teacher to generate the appropriate demonstration as done in Caselles-Dupré et al. (2022), rather than selecting demonstrations from a provided set. Finally, the prior information introduced in the teacher's Bayesian ToM model of the learners, particularly through belief supports, could be reduced by employing deep neural network-based ToM models as in Rabinowitz et al. (2018).

ACKNOWLEDGEMENTS

Anonymized for review.

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

# A  TOY ENVIRONMENT

We test our ToM-teacher on a toy environment, see Figure 4. This simpler environment is another instance of the formal framework presented in the main paper. We consider POMDPs instead of GC-POMDPs, where the learner is characterised solely by a belief rather than a goal and an observation function. This reduces the support of ToM-teacher's beliefs but the general framework remains unchanged.

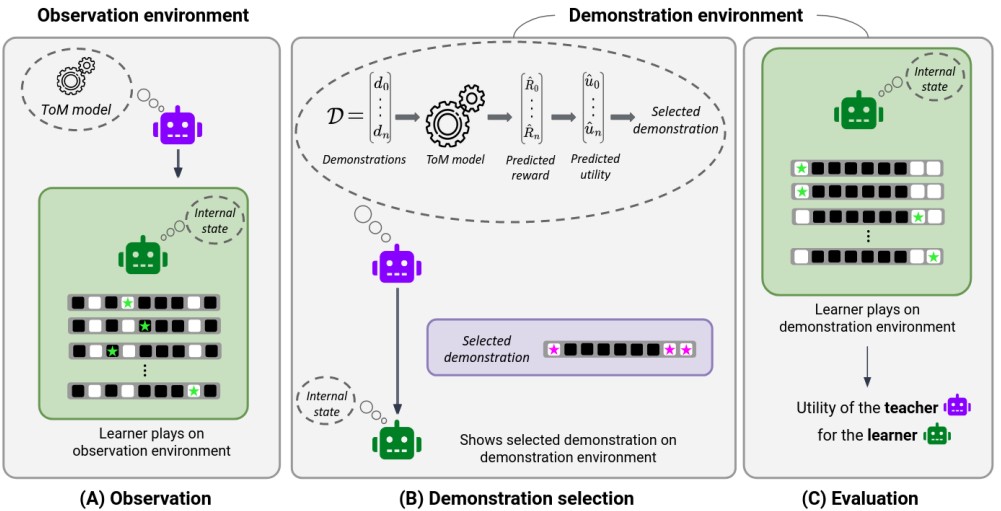

**(A) Observation**          **(B) Demonstration selection**          **(C) Evaluation**

Figure 4: Interaction with toy environments, where white cells represent musical buttons, black cells represent silent buttons, and a star represents a button press. (A) The teacher 🤖 observes a learner 🤖 with a particular internal state behaving in a simple environment $\mathcal{M}^{\text{obs}}$ and infers a ToM model of this learner. (B) In a more complex environment $\mathcal{M}^{\text{demo}}$, the teacher uses this ToM model to predict the usefulness for the observed learner of each demonstration of a provided dataset $\mathcal{D}$, out of which it selects the utility-optimal demonstration $d^*$. The learner observes $d^*$ and updates its knowledge about $\mathcal{M}^{\text{demo}}$. (C) The learner behaves in $\mathcal{M}^{\text{demo}}$ and receives a reward. The teacher is evaluated on the utility of $d^*$, which is the learner's reward minus the cost incurred by the teacher in delivering that demonstration.

## A.1  ENVIRONMENT

The environment is reminiscent to the toy used in the study presented in Gweon et al. (2018). It represents a toy with $N = 20$ buttons among which $M = 3$ buttons produce music and the rest do nothing. As mentioned earlier, this environment is formalised as a POMDP $\mathcal{M}^j = (\mathcal{S}^j, \mathcal{A}, \mathcal{T}^j, R^j)$. We model it as a 1D gridworld with a single state $\mathcal{S}^j = \{s^j\}$ with $s^j[n] = 1$ if the $n^{th}$ button produces music an $s^j[n] = 0$ otherwise, and identity transition function. The action space is the set of buttons $\mathcal{A} = [0, N-1]$. Unlike in GC-POMDPs, where the reward function is defined over trajectories, in this context, the reward function is defined for individual actions. Specifically, the reward function is $\forall a \in \mathcal{A}, \quad R^j(a) = s^j[a]$. Contrary to the main paper, in this environment, we consider that all the agents share the same observation function that reveals the state of one cell at a time, $v(a) = s^j[a]$.

One trajectory of an agent in $\mathcal{M}^j$ is defined by the action-reward pairs $\{(a_k^j, r_k^j = s^j[a_k^j])\}_{k=0}^{K-1}$. As mentioned in Section 3.1, we consider environments $\mathcal{M}^j$ sharing the same action space. This means that we consider toys with the same number of buttons but different musical ones.

## A.2  LEARNER

As defined in Section 3.2, the learner has a belief on the state of the environment and updates it from observations following the Bayesian update rule Equation 1.

However, in this particular case, the conditional probability of the observation $o_t = v(a_t) = s^j[a_t]$ knowing the state of the environment $s \in \mathcal{S}^j$ is $\mathbb{P}(o_t|s) = \mathbb{1}\left(o_t = s[a_t]\right)$.

In contrast to 2D navigation task, where learners were defined both by their goals and beliefs (i.e., observation functions), in this environment, we characterise learners solely by their beliefs. Additionally, these beliefs do not vary because of different observation functions but because of their initial values $b_0^L$.

For a learner, the initial belief corresponds to prior knowledge on the environment. In the context of the study presented in Section 1 from Gweon et al. (2018), if a child has previously interacted with a toy featuring a single musical button ($M = 1$), their initial belief about a similar new toy would be that it also has one musical button. As a result, we can assume that their initial belief only assigns nonzero probability to the configurations of the toy that contain exactly one musical button. Similar reasoning can be extended to the prior beliefs corresponding to toys with two or three musical buttons. The initial belief of each learner is expressed as:

$$\forall i \in [1,2,3], \quad \forall s \in \mathcal{S}_B^j, \quad b_0^{L_i}(s) = \frac{\mathbb{1}\left(\left(\sum_n^{N-1} s[n]\right) = i\right)}{\sum_{s' \in \mathcal{S}_B^j}\left(\left(\sum_{n'}^{N-1} s'[n']\right) = i\right)} = \frac{\mathbb{1}\left(\left(\sum_n^{N-1} s[n]\right) = i\right)}{\binom{N}{i}}.$$

The last learner $L_0$ does not have any prior on the environment:

$$\forall s \in \mathcal{S}_B^j, \quad b_0^{L_0}(s) = \frac{1}{|\mathcal{S}_B^j|} = \frac{1}{2^N}.$$

All the learners follow a policy conditioned by their beliefs: if they are certain about the state of the environment they play greedy, otherwise they explore. This policy aligns with the concept underlying the learner's policy in the 2D navigation task, but in this case, it is formalised as follows:

$$\forall a \in \mathcal{A}^j, \quad \pi(a|b^L) = \begin{cases} \frac{\mathbb{1}(s[a]=1)}{\sum_{a' \in \mathcal{A}^j}(s[a']=1)} & \text{if } \exists s \in \mathcal{S}_B^j \text{ s.t } b^L(s) = 1 \\ \frac{1}{|\mathcal{A}^j|} & \text{otherwise.} \end{cases} \quad (7)$$

A learner $L_i$ is thus entirely defined by its initial belief and belief-conditioned policy, $L_i = (\pi, b_0^{L_i})$.

## A.3 TEACHER

### A.3.1 UTILITY BASED DEMONSTRATION SELECTION STRATEGY

In this experiment, the set of demonstrations includes examples of one, two, or three musical buttons, along with a demonstration revealing all the buttons of the toy. Intuitively, to achieve maximal reward on $\mathcal{M}^{\text{demo}}$, a learner has to play greedy, thus has to be certain about the toy's configuration, see Equation 7. Therefore, a learner $L_i$ exposed to a demonstration revealing $i$ or more buttons can achieve maximal reward.

### A.3.2 TOM-TEACHER

The ToM-teacher is identical to the one defined in Section 3.3.3. The teacher has to help an unknown learner $L$ by providing it with a demonstration. It has a belief $b^T$ on the internal state of the learner and has access to an approximation of the learner policy $\hat{\pi}$ as well as to one of its past trajectory $\tau^{\text{obs}} = \{(a_k^{\text{obs}}, r_k)\}_{k=0}^{K-1}$, on an environment $\mathcal{M}^{\text{obs}}$. In this experiment, the learners are characterised solely by their belief (i.e., initial belief). Therefore, the support of the ToM-teacher's belief is a finite set of possible initial beliefs $\mathcal{B}_B$. In this context, Equation 4 can be simplified by:

$$\forall b_0^i \in \mathcal{B}_B, \quad b_{k+1}^T(b_0^i) = \frac{b_k^T(b_0^i) \times \hat{\pi}(a_k^{\text{obs}}|b_k^i)}{\sum_{b_0^{i'} \in \mathcal{B}_B} b_k^T(b_0^{i'}) \times \hat{\pi}(a_k^{\text{obs}}|b_k^{i'})},$$

with $b_k^i$ the belief that would have learner $L_i = (\hat{\pi}, b_0^i)$ after having observed the slice of the trajectory $\tau^{\text{obs}}[0:k-1]$.

The remainder of the interaction is consistent with the main paper: after the teacher updates its belief, it estimates the utility of each possible demonstration and chooses the one that maximises this estimation.

### A.4 EXPERIMENTS

**Environment:** The observation environment $\mathcal{M}^{\text{obs}}$ is a toy with $N = 20$ buttons, including $M = 3$ musical buttons that are randomly distributed on the toy. The demonstration environment $\mathcal{M}^{\text{demo}}$ features the same toy but with different locations for the musical buttons.

**Learner:** We consider learners with four different initial beliefs, see Section A.2, and sharing the policy described in Equation 7.

**Demonstration set:** The demonstration set contains four samples: three demonstrating one, two and three musical buttons respectively, and one demonstration featuring all buttons. We use a linear teaching cost $c_\alpha(d) = \alpha \times len(d)$ with $\alpha = 0.03$, meaning the longest demonstration of size 20 has a cost of 0.6.

**Teacher:** For the experiments, we only consider an ■ *aligned ToM-teacher*, with access to the true policy of the learners, $\hat{\pi} = \pi$. We use the same baselines as in the main paper (see Section 4): the ■ *omniscient teacher* providing an upper bound on the teacher's utilities, the ■ *utility-optimal non-adaptive teacher*, the ■ *reward-optimal non-adaptive learner*, and the ■ *uniform sampling teacher*. In these experiments, all demonstrations are specific to each learner, resulting in the fusion of the *uniform modelling* and *uniform sampling* teachers.

**Metrics:** The utility of the teacher is computed on one action $a_L$ of the learner on $\mathcal{M}^{\text{demo}}$ after having observed the demonstration $d^*$ selected by the teacher: $u_\alpha^{\text{demo}}(d^*, L) = R^{\text{demo}}(a_L) - c_\alpha(d^*)$.

**Experiments**: We conducted 300 experiments for each learner. In each trial, both $\mathcal{M}^{\text{obs}}$ and $\mathcal{M}^{\text{demo}}$ are randomly generated, and all teachers are evaluated within the same environment pair $(\mathcal{M}^{\text{obs}}, \mathcal{M}^{\text{demo}})$ – all teachers select a demonstration from the same demonstration set $\mathcal{D}$.

### A.5 RESULTS

Figure 5 illustrates the mean utility of the demonstrations selected by each teacher, for learners with varying initial belief on $\mathcal{M}^{\text{obs}}$ observed for 40 actions.

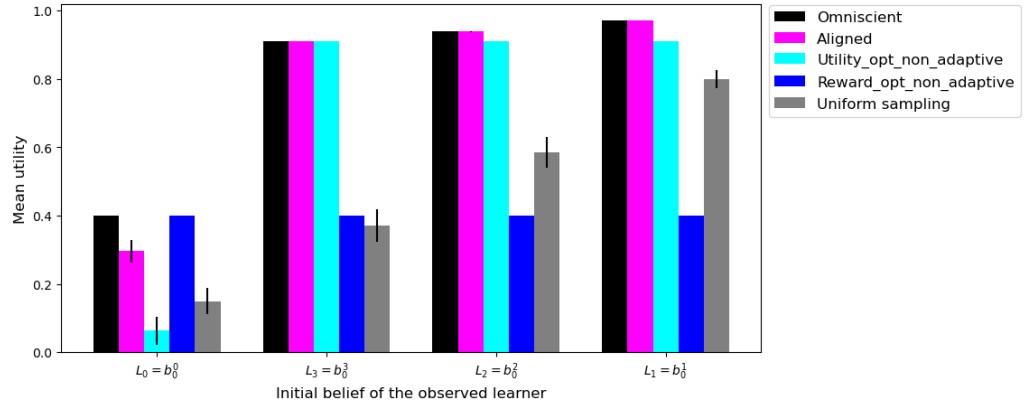

Figure 5: Mean utilities and 95% confidence interval of the aligned ToM-teacher and baseline teachers for learners with varying initial beliefs observed on $\mathcal{M}^{\text{obs}}$ during 40 actions.

Similarly to the results found in the 2D navigation task (Section 5), the aligned ToM-teacher outperforms the learner-agnostic teachers for learners who believe the toy contains one or two musical buttons (p-values $< 0.01$). Furthermore, the utility of the demonstrations provided by the aligned ToM-teacher for learners who believe there are one, two, or three musical buttons, shows no significant difference from that of the omniscient teacher (p-values $> 0.9$).

However, in this experiment, since all the provided demonstrations are optimal for specific learners, the reward-optimal non-adaptive teacher, which provides a demonstration featuring all the buttons of the toy, performs not significantly differently from the omniscient teacher for the learners for whom this demonstration is optimal (p-value $> 0.9$), i.e., the learners with no initial prior knowledge about the toy. Similarly, with a teaching cost parameter of $\alpha = 0.03$, the most useful demonstration, on average across all learners, is the one displaying the three musical buttons. Therefore, the demonstration selected by the utility-optimal non-adaptive teacher achieves mean utility not significantly different from that of the aligned ToM-teacher and omniscient teacher for learners requiring to be shown exactly three musical buttons (p-values $> 0.9$).

Note that in this particular environment, there are cases in which both learners with no prior knowledge about the environment and learners believing there are three musical buttons behave the same. Therefore, the teacher's belief about the learner's internal state cannot exceed $0.5$. With a high teaching cost parameter of $\alpha = 0.03$, the teacher selects the less costly demonstration, resulting in the poorer utility of the aligned ToM-teacher for learners with uniform initial beliefs, requiring the display of the entire toy.

In this environment, the aligned ToM-teacher achieves high utility for all learners, while the learner-agnostic teacher selects demonstrations that are beneficial to some learners but poorly useful to others. Just as in the 2D navigation task, modelling the learner's internal state, which in this case represents an initial prior belief about the toy, proves to be essential for selecting useful demonstrations for all the learners under consideration.

## B  IMPLEMENTATION DETAILS

All the code necessary to reproduce the experiments from the main paper, the appendix, and generate the figures can be found at `https://github.com/teacher-with-ToM/Utility_Based_Adaptive_Teaching`.

### B.1  LEARNERS' POLICY

The learner always starts at the bottom centre of the environment, i.e. at coordinates $(s - 1, \lfloor \frac{s}{2} \rfloor)$ when dealing with an environment of size $s \times s$.

In cases where the view is partial, the learner's sight is bounded by the walls, as defined in the `MiniGrid` library.

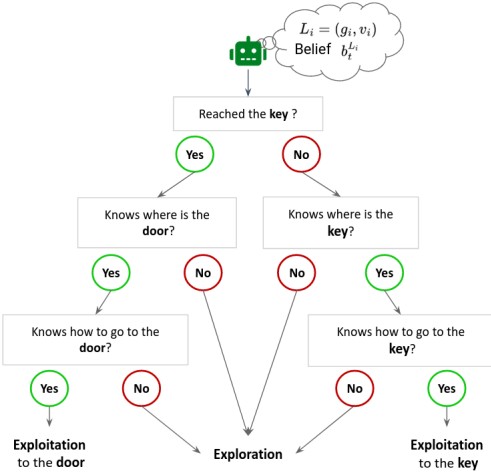

Figure 6: Policy 2 of the learner conditioned by its goal, observation function and belief for 2D navigation task

We construct the learner's policy as a decision tree, as illustrated in Figure 6. We assume that the learner first searches for the key before heading to the door. Similar to a Bayes-optimal policy (Duff & Barto, 2002), the learner leverages its uncertainty about the environment to select its actions.

**Exploitation:** Specifically, when the learner knows the precise location of either the key or the door, it follows a greedy policy, following the shortest path computed by the $A^*$ algorithm with a Manhattan distance heuristic. This information is contained in the learner's belief.

**Exploration:** Uncertainty also plays a crucial role in actively exploring the environment when the learner lacks information about the object's location that it aims to reach. At time $t$, the learner selects action $a_t$ to maximally reduce its uncertainty. In other words, the learner chooses at time $t$ an action $a_t$ that will provide the greatest amount of new knowledge about the environment at time $t + 1$. Therefore, we can derive $a_t$ from the following equation: $a_t = \arg\max_{a \in \mathcal{A}} \{\sum_{c \in v(s')} H(b_t^L[c]) \text{ s.t } s' = \mathcal{T}(s, a)\}$, where $H$ is the Shannon entropy[2] and $b_t^L[c]$ the belief of the learner about the cell $c$ of the MiniGrid environment.

## B.2 Demonstration

To generate the demonstrations we utilise the Nearest Neighbour Algorithm based on distance maps computed by the Dijkstra algorithm and following shortest paths computed by the $A^*$ algorithm with the Manhattan distance heuristic.

To generate a demonstration, we define a set of objects to be shown for a receptive field size. The first object to be shown is the one closest to the initial agent's position based on the Dijkstra distance map. The agent follows the shortest path computed by the $A^*$ algorithm until the object appears in the receptive field. Then, we repeat the process, moving to the object closest to the current agent's position, until all objects have been shown.

Specifically, for learner-specific demonstrations, the set of objects includes the key and door of the learner's goal colour, and the receptive field size matches that of the learner. For unspecific demonstrations, objects are randomly selected from the set of objects present in the environment, and we use the smallest receptive field of size 3.

## B.3 Rational ToM-teacher

The model of this teacher is based on two assumptions (1) the learner will grab the key before heading to the door (2) The learner is rational, meaning it will try to open the door in the minimal amount of time.

These assumptions give rise to an approximation of the learner's policy $\hat{\pi}$ conditioned by a belief, an observation function and a goal. Similarly to the real learner's policy, $\hat{\pi}$ can be decomposed in an exploitation policy when the the learner knows the location of the object it want to reach and an exploration policy otherwise.

**Exploitation:** We model the learner's level of rationality using a Boltzmann policy that favours actions bringing the learner closer to the object it needs to reach.

In MiniGrid environments, the learner has a position and an orientation, and the $left$ and $right$ actions only change the learner's orientation. As a result, when taking these actions, the spatial distance between the learner and the object remains identical, leading to the contradiction that 'going left then forward' is improbable compared to 'going forward then left'. Therefore, when following a greedy strategy, we assume that if the learner selects the $left$ (resp. $right$) action, it intends either to perform a u-turn (going to the position behind itself) or to move left (resp. right). The learner's approximate exploitation policy, when attempting to reach an object $obj$ for which it is certain about the location $p_{\text{obs}}$, i.e., $\sum_{\{s \,:\, s[p_{obj}]=obj\}} b^L(s) = 1$, is expressed as follows:

---

[2]The Shannon Entropy of a probability distribution $P$ with discrete support is $H(P) = -\sum_{x \in supp(P)} P(x) \times \log(P(x))$.

$$\hat{\pi}(s, a|g, v, b^L) = \begin{cases} \frac{\exp\left(-\left(d(p_{t+1}^{u\_turn}, g) - d(p_t, p_{\text{obs}})\right)/\lambda\right) + \exp\left(-\left(d(p_{t+1}^a, p_{\text{obs}}) - d(p_t, g)\right)/\lambda\right)}{2 \times Const} & \text{if } a \in \{left, right\} \\ \frac{\exp\left(-\left(d(p_{t+1}^a, g) - d(p_t, p_{\text{obs}})\right)/\lambda\right)}{Const} & \text{if } a = forward, \end{cases}$$

where $p_t$ represents the current position of the learner, $p_{t+1}^{u\_turn}$ is the resulting position after performing a u-turn, and $p_{t+1}^a$ is the position after taking action $a$ if $a$ is $forward$, and taking action $a$ and going $forward$ is $a \in \{left, right\}$. The distance $d$ between two objects in the environment is computed using Dijkstra's algorithm.

For $\lambda \to 0$, $\hat{\pi}$ models a learner following a shortest path towards the object. Conversely, for $\lambda \to \infty$, the approximate policy models a learner uniformly selecting an action, thus $\lambda$ reflects the learner's degree of rationality.

**Exploration:** This teacher does not have any prior knowledge about the exploration policy and assumes it to be uniform.

In our experiment in Section 4, the learners' policy is based on a shortest path algorithm, representing perfect rationality. Therfore, we employ the lowest temperature parameter $\lambda$ for the rational ToM-teacher. However, if the learner's policy were learned, the temperature parameter would also need to be learned to align with the learner's policy which might be noisily rational.

## C    ADDITIONAL RESULTS

Here we present the precise values along with 95% confidence interval of the mean utilities of the ToM and baseline teachers in the experiments presented in the main paper and displayed in Figure 2 when the teacher has access to a full trajectory of the learner and Figure 3 when the teacher's observation of the learner is limited to 10 actions.

Table 1: Mean utilities and 95% confidence interval of the demonstrations selected by the ToM-teachers and the baseline teachers after having observed the learners on a full episode. The rational teacher has a temperature parameter of $\lambda = 0.01$.

| Teacher | Utility for learners with receptive field size of 3 | Utility for learners with receptive field size of 5 | Utility for learners with full observability |
|---|---|---|---|
| Aligned ToM-teacher | $0.64 \pm 0.01$ | $0.68 \pm 0.01$ | $0.86 \pm 0.01$ |
| Rational ToM-teacher | $0.54 \pm 0.03$ | $0.62 \pm 0.02$ | $0.85 \pm 0.01$ |
| | | | |
| Omniscient | $0.65 \pm 0.01$ | $0.68 \pm 0.01$ | $0.90 \pm 0.00$ |
| Utility-optimal non-adaptive | $0.14 \pm 0.03$ | $0.33 \pm 0.03$ | $0.79 \pm 0.01$ |
| Reward-optimal non-adaptive | $0.28 \pm 0.01$ | $0.28 \pm 0.01$ | $0.31 \pm 0.00$ |
| Uniform sampling | $0.01 \pm 0.03$ | $0.15 \pm 0.04$ | $0.60 \pm 0.02$ |
| Uniform modelling | $0.01 \pm 0.03$ | $0.17 \pm 0.04$ | $0.76 \pm 0.01$ |

## D    INTERNAL STATE INFERENCE

In order to evaluate the ToM models of our teachers, we derive explicit inferences of the learner's goal and receptive field size by taking the Maximum A Posteriori (MAP) estimations on the teacher's belief. At step $k$ of the observed learner's trajectory, the teacher's belief about the learner's goal is the marginal probability $\forall g \in \mathcal{G}_B, \quad b_k^T(g) = \sum_{v \in \mathcal{V}_B} b_k^T(g, v)$, —a similar formulation applies to the teacher's belief regarding the learner's receptive field size. In that sense, the MAP estimator

Table 2: Mean utilities and 95% confidence interval of the demonstrations selected by the ToM-teachers and the baseline teachers after having observed the learners on the 10 first steps of an episode. The rational teacher has a temperature parameter of $\lambda = 0.01$. In **bold**, the mean utilities that are significantly different from the first experiment Table 1 (p-values $< 0.01$).

| Teacher | Utility for learners with receptive field size of 3 | Utility for learners with receptive field size of 5 | Utility for learners with full observability |
|---|---|---|---|
| Aligned ToM-teacher | **0.39 ± 0.02** | **0.62 ± 0.02** | 0.86 ± 0.01 |
| Rational ToM-teacher | **-0.03 ± 0.04** | **0.14 ± 0.05** | **0.63 ± 0.04** |
| | | | |
| Omniscient | 0.64 ± 0.01 | 0.70 ± 0.01 | 0.90 ± 0.00 |
| Utility-optimal non-adaptive | 0.13 ± 0.03 | 0.36 ± 0.03 | 0.79 ± 0.01 |
| Reward-optimal non-adaptive | 0.27 ± 0.01 | 0.29 ± 0.01 | 0.31 ± 0.00 |
| Uniform sampling | 0.00 ± 0.03 | 0.20 ± 0.04 | 0.58 ± 0.02 |
| Uniform modelling | −0.01 ± 0.03 | 0.18 ± 0.04 | 0.76 ± 0.01 |

of the learner's goal is $\hat{g}_k^T = \arg\max_{g \in \mathcal{G}_B} b_k^T(g)$, and equivalently for the MAP estimation of the learner's receptive field size $\hat{v}_k^T$.

In Figure 7, the ToM models are evaluated on the accuracy of their MAP estimators as well as on their uncertainty. If learner $L_i$ is observed on $k$ steps of its trajectory, the accuracy of the teacher's MAP estimators of the learner's goal (*Goal-inference accuracy*), and receptive field size (*RF-inference accuracy*) are $\mathbb{1}(g_i = \hat{g}_k^T)$ and $\mathbb{1}(v_i = \hat{v}_k^T)$ respectively. We also evaluate the uncertainty of the ToM models with the Shannon entropy of the teachers' beliefs. The higher the Shannon entropy, the higher the teacher's uncertainty about the goal and receptive field size of the learner.

As soon as the learner grabs the key, the teacher is certain about the learner's goal. We define a first phase of the episode corresponding to the segment of the trajectory before the learner obtains the key. We evaluate the accuracy and uncertainty of the ToM models about the learner's goal along this first phase and the accuracy and uncertainty about the learner's receptive field size along the entire episode.

We compare the MAP estimators and uncertainty of aligned and rational ToM-teachers with varying temperature parameter $\lambda \in [0.01, 0.5, 1., 3., 10]$ – the lower the temperature parameter, the more rational the learner is assumed to be.

**Goal-inference:**

The MAP estimators of the learner's goal perform not significantly different for all teachers at both the very beginning and the end of the first phase of the episode. In fact, all teachers start with uniform beliefs, resulting in uniform MAP estimators and achieve perfect accuracy as soon as the learner grabs the key. Nonetheless, at every stage, the accuracy of the teachers' MAP estimators of the learner's goal is significantly higher for the teachers with more accurate behavioural models of the learner, i.e. aligned teachers and rational teachers with low-temperature parameters.

We can observe that while the accuracy of all MAP estimators increases throughout the first phase of the episode, the beliefs about the learner's goal of the rational ToM-teachers with extreme values of the temperature parameter ($\lambda = 3$ and $\lambda = 10$) remain weak: the uncertainty remains high and abruptly decreases when the learner obtains the key – at the end of the first phase of the episode. In contrast, the aligned and rational ToM-teachers with low temperature parameters exhibit both linear increase of the MAP estimator accuracy and decrease in uncertainty throughout the first phase. This demonstrates their ability to accurately and confidently infer the goal based on the learner's behaviour before acquiring its key.

**RF-inference:** Inferring the receptive field size of the learner is a more difficult task than inferring its goal. As a consequence, having a model of the learner's policy proves to be essential to correctly

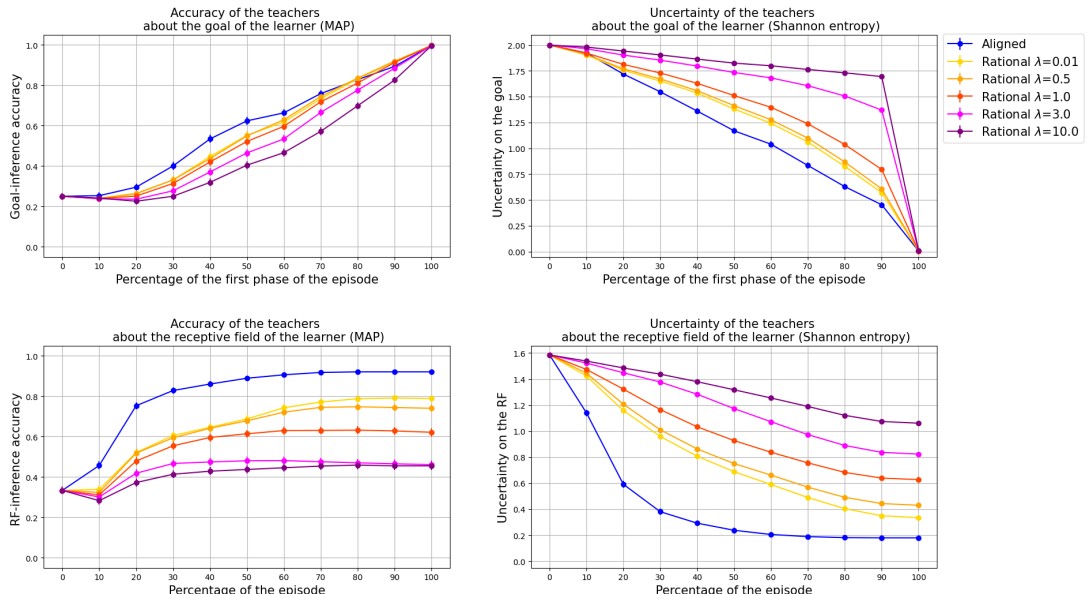

Figure 7: Mean and $95\%$ confidence interval of the accuracy of the MAP estimators and the uncertainty (Shannon entropy of the teachers' beliefs) of the aligned and rational ToM-teachers with varying temperature parameter $\lambda$, regarding the learner's goal and receptive field size respectively as a function of the percentage of the first phase and of the entire episode.

draw inferences from the learner's behaviour. The MAP estimator of the learner's receptive field size of the aligned teacher converges faster in accuracy ($\approx 60\%$ of the episode) and reaches an accuracy superior to $0.9$ while all rational teachers achieve lower accuracy on average at the end of the episode and need more observation of the learner to converge (for the teacher with higher assumed level of rationality, $\lambda = 0.01$, the accuracy of the MAP estimator converges at $\approx 80\%$ of the episode towards an accuracy of $\approx 0.8$).

At any given stage in the episode, the accuracy and the confidence of the ToM model on the learner's internal state are inversely proportional to the accuracy of the teacher's behavioural model of the learner: the less accurate the model, the more inaccurate (lower accuracy) and uncertain (higher Shannon entropy) the inferences are. In the context of limited observation of the learner (i.e., early stages of the episode) this explains the results obtained in Section 5.2.

# E    EXAMPLE OF MISREADING LEARNER'S BEHAVIOUR

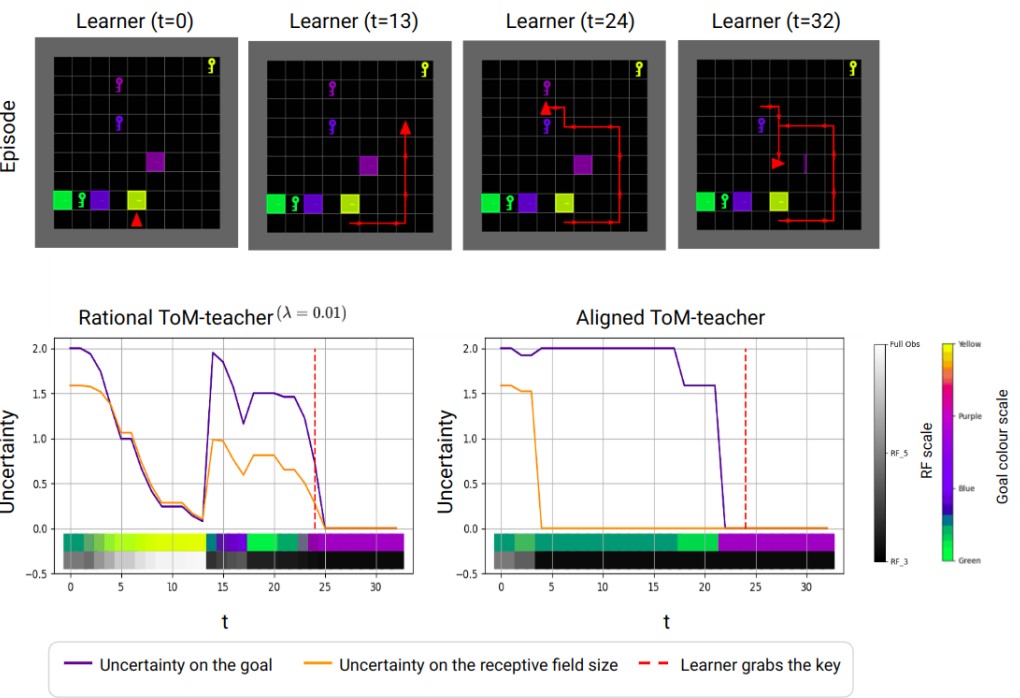

Figure 8: Evolution of uncertainty regarding the learner's goal and the receptive field size for both the rational ToM-teacher ($\lambda = 0.01$) and the aligned ToM-teacher during the observation of a learner (Figure 1 (A)) with a real goal of *purple* and a receptive field size of 3. At every time $t$, the colour bars represent the average goal colour and receptive field size weighted by the teachers' beliefs.

Figure 8 illustrates a case in which, during the episode, the rational ToM-teacher misinterprets the learner's behaviour, drawing false conclusions about its internal state, while the aligned ToM-teacher avoids such mistakes, having greater insight into the learner's behavioural policy. In fact, at the beginning of the trajectory, while exploring, the learner gets closer to the yellow key. Consequently, the rational ToM-teacher believes that the learner's goal colour is *yellow* and it has full observability. On the other hand, the aligned ToM-teacher recognises that the learner is not following the path it would have taken if it indeed had the assumed internal state, thereby avoiding any misinterpretation.

In this example, if the teachers' observation of the learner had been limited to only 10 actions, the rational ToM-teacher would have erroneously selected the demonstration for a learner with a *yellow* goal and full observability, resulting in poor utility on the more complex demonstration environment. By contrast, the aligned ToM-teacher would have selected the demonstration maximising the mean utility for learners with different goals but a receptive field of size 3, as it would be certain about the sensory capacity but not the goal. This example illustrates the results found in Section 5.2 as well as in Appendix D.

# F    TEACHING COST PARAMETER

With a low teaching cost parameter of $\alpha = 0.1$, the utility-optimal teacher selects the same demonstrations as the reward-optimal teacher, specifically the demonstration showing all objects in the environment. As a result, they achieve utilities that are not significantly different for all the learners (p-values $> 0.6$). These teachers' utilities approach those of the ToM and omniscient teachers. Furthermore, the utilities of the ToM teachers are closer to those of the omniscient teacher because making a mistake becomes less costly. In the context of a low teaching cost, modelling the internal state of the learner makes little difference. Specifically, with a teaching cost of $\alpha = 0$, all teachers except uniform ones perform the same, matching the omniscient teacher's utility.

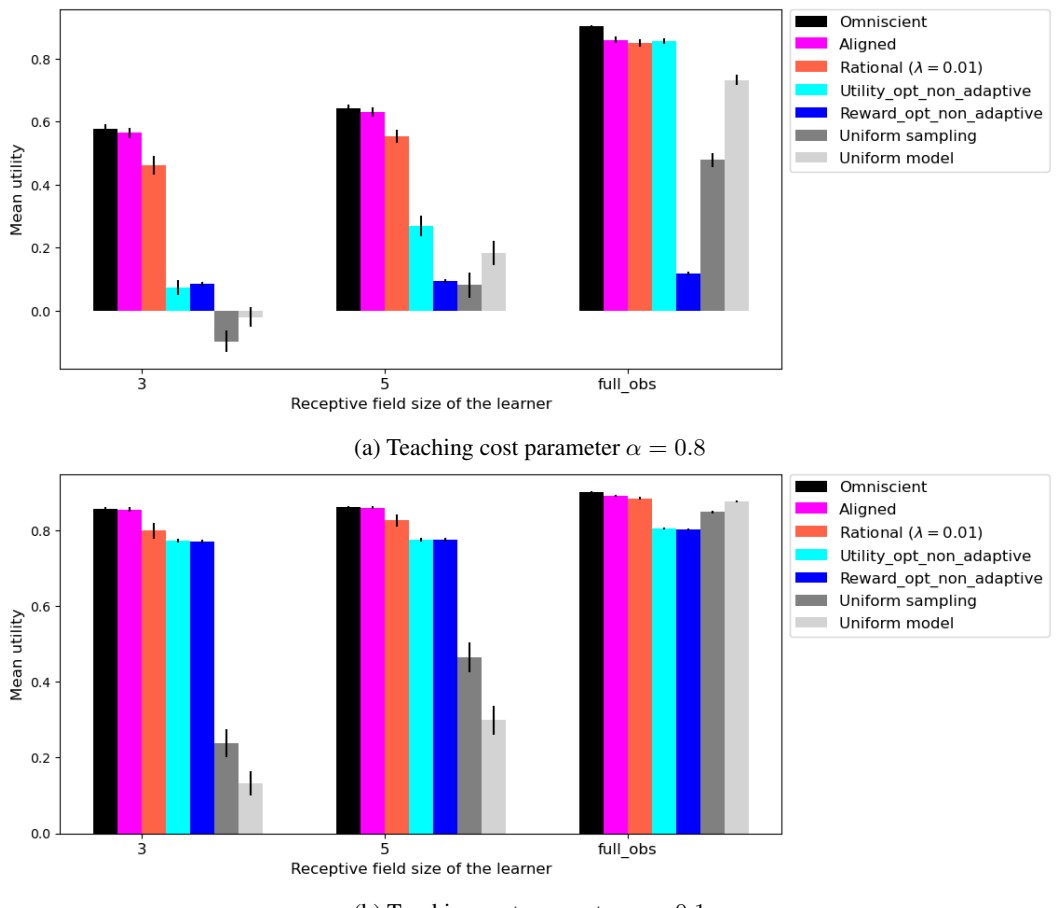

(a) Teaching cost parameter $\alpha = 0.8$

(b) Teaching cost parameter $\alpha = 0.1$

Figure 9: Mean utilities and 95% confidence interval of ToM-teachers (rational teacher with parameter $\lambda = 0.01$) and baseline teachers for learners with varying receptive field sizes of $[3, 5, full\_obs]$ observed on $\mathcal{M}^{\text{obs}}$ during a full episode.

On the other hand, with a high teaching cost parameter of $\alpha = 0.8$, the reward-optimal demonstration showing all the objects in the environment becomes too expensive and leads to poor utility for all learners. The utility-optimal teacher selects less informative demonstrations with low teaching cost leading to high mean utility not significantly different (p-values $> 0.4$) from that of the ToM teachers, for learners that require no information (i.e. learners with full observability). However, this strategy results in poor utilities for learners with limited receptive field size, who require informative and specific demonstrations to achieve their goal in the complex environment. Therefore, with a high teaching cost, modelling the learner's internal state becomes essential to select useful demonstrations for all learners.

## G   SIZE OF THE OBSERVATION ENVIRONMENT

In our framework, the observational environment $\mathcal{M}^{\text{obs}}$ must be simple enough for all learners to interact with the objects, enabling the ToM-teacher to infer their mental states. On the contrary, the demonstration environment $\mathcal{M}^{\text{demo}}$ where the teacher conducts teaching needs enough complexity for learners to require the teacher's assistance to achieve the task.

However, while maintaining this requirement, the complexity of the observation environment can vary. To evaluate how this complexity influences the accuracy of the teachers' ToM models, we compute the accuracy of the MAP estimators for the learner's goal and receptive field, as defined in Appendix D. This evaluation occurs after observing a single trajectory of the learner in observation

environments of different sizes. To ensure that all learners can interact with the objects, we employ the same MiniGrid environment defined in Section 4, but within an environment of size $s \times s$ with $s \in [11, 15, 21, 25]$, where the maximum number of steps is set to max_steps $= s^2$.

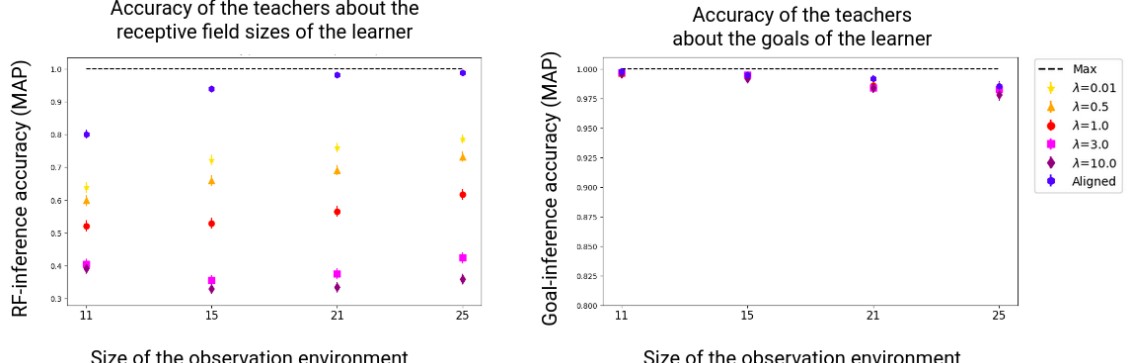

Figure 10: Mean and $95\%$ confidence interval of the accuracy of the MAP estimators of the aligned and rational ToM-teachers with varying temperature parameter $\lambda$, regarding the learner's goal and receptive field size respectively, as a function of the size of the observation environment $\mathcal{M}^{\mathrm{obs}}$.

The larger the environment's size ($s \times s$), the greater the dispersion of its objects. As indicated in Figure 10, the increment in environment size has only a minor impact on the accuracy of the MAP estimator for the ToM models concerning the learners' goals. Specifically, in scenarios devoid of obstacles (apart from the doors and keys) and with sufficient timesteps, all learners can access their respective keys and/or doors, resulting in high goal accuracy of all the teachers. Nonetheless, a slight reduction in accuracy is observed in larger environments. Occasionally, even within expansive environments lacking inner walls, certain learners may be unable to reach their keys and doors. The limited interaction with the objects hampers the ToM model's ability to accurately infer their goals.

Additionally, in Figure 10, we note that the accuracy of the MAP estimator for the receptive field sizes of the learners, specifically for ToM-teachers equipped with an accurate model of the learners' policy (teachers with low temperature parameter $\lambda$ or aligned), increases with the environment's size. This outcome can be attributed to two phenomena. Firstly, as learners navigate larger environments, they undertake a greater number of actions to reach their goals. Consequently, the teacher gains access to longer sequences of observations, enabling a more refined modeling of the learner's internal state. Secondly, the increased dispersion of objects within larger environments reduces the ambiguity between the behaviours of learners with different sizes of receptive field, thereby resulting in higher accuracy within larger environments.

