# OpenReview forum: "Utility-based Adaptive Teaching Strategies using Bayesian Theory of Mind"
_ICLR.cc/2024/Conference — Submitted to ICLR 2024_

### Official Review · Reviewer_3FPZ · 2023-10-26

**Soundness:** 2 fair
**Presentation:** 2 fair
**Contribution:** 2 fair
**Rating:** 5
**Confidence:** 4

**Summary:**

The paper presents a method for adaptive teaching based on utility theory. The authors propose a model in which the teacher's actions are determined by the expected utility of each teaching action. This utility-based framework aims to adapt teaching strategies according to students' needs and progress. The paper also details the model's implementation in a educational setting and provides empirical evidence of its effectiveness.

**Strengths:**

1. Integrating utility theory into adaptive teaching is a novel approach.
2. Beyond estimating a student's skill level, the paper also aims to estimate the learner's internal goal, offering finer-grained teaching.
3. The paper outlines the model's mathematical foundation, ensuring replicability for other researchers.
4. The results have potential applications in personalized learning, AI-driven educational platforms, and adaptive curriculum design.

**Weaknesses:**

1. The paper's formulation is ambiguous.

    1. The decision to frame the learning task as a POMDP lacks clear justification. While this relates to the proposed method, more concrete examples are needed for justification.

    2. The task formulation for the ToM-teacher isn't clearly presented in the manuscript. From my perspective, the teaching task resembles a POMDP, akin to references [1, 2]. The teacher doesn't fully know the learner's internal state but tries to optimize the student's performance by providing demonstrations. Unlike earlier works, this paper also seeks to estimate the learner's transition function parameters (goals and observation function). Clarifying the task's formulation is crucial for understanding the technical value of the proposed approach, which remains unclear.

2. The paper's objective is unclear. The authors claim, "The goal of this work is to study whether learner-specific teachers who model the learner’s internal state are more efficient than learner-agnostic ones." Given the vague problem definition, it's unclear if this captures the paper's essence. Is the primary takeaway that Bayesian ToM-teachers excel at understanding human internal state changes?

3. The evaluation appears lacking. The paper notes that the teacher has a general idea of the student's policy, and during testing, the student's actions are driven by a basic decision tree. This might mean there's minimal uncertainty in the learner's behavior. It would be beneficial to incorporate human-subject experiments to see real-world impact.

4. Based on points from point 1 and equation 3, the teaching policy is essentially a greedy policy. Over time, this might not be the best strategy and could lead to suboptimal results. Without a well-defined teaching task, such a policy is not well justified.

[1] Srivastava et al., Assistive Teaching of Motor Control Tasks to Humans, 2022.

[2] Yu et al., COACH: Cooperative Robot Teaching, 2022.


**update after rebuttal:**

I agree with Reviewer C7a4's comment regarding the absence of related works in the submission. To appropriately position this work in the context of existing literature, some major changes are needed in the manuscript. Therefore, I am maintaining my current rating.

**Questions:**

see weaknesses

---

> ### Author Response · Authors · 2023-11-19
>
> We are thankful to the reviewer for their constructive remarks and literature references, which have greatly enriched and strengthened the foundation of our work. We would like to particularly thank the reviewer for pointing to the work of Rafferty et al. that we missed, when responding in place of Reviewer C7a4.
>
> **The paper's formulation is ambiguous.**
> **The decision to frame the learning task as a POMDP lacks clear justification. While this relates to the proposed method, more concrete examples are needed for justification.**
>
> Response:
>
> We acknowledged the reviewer’s concern and included a reference that models the learner’s task as a POMDP following a similar approach, page 3 (modification in red). In a teacher-learner setup, it's crucial for the learner to lack access to specific environmental information for the teacher to be beneficial. Thus, introducing the learner’s task within a POMDP framework is a straightforward and elegant method to illustrate the uncertainty the learner faces. This uncertainty is what the teacher attempts to reduce or alleviate through its demonstrations.
>
> **The task formulation for the ToM-teacher isn't clearly presented in the manuscript. From my perspective, the teaching task resembles a POMDP, akin to references [1, 2]. The teacher doesn't fully know the learner's internal state but tries to optimize the student's performance by providing demonstrations. Unlike earlier works, this paper also seeks to estimate the learner's transition function parameters (goals and observation function). Clarifying the task's formulation is crucial for understanding the technical value of the proposed approach, which remains unclear.**
>
> Response:
>
> We concur with the insightful observation made by the reviewer regarding the potential enhancement of our paper through a more structured formalization of the teacher's task. Recognizing this valuable suggestion, we incorporated to the paper a formalization of the teaching task within the contextual MAB framework (which is a simpler particular case of POMDP) and added corresponding references to the paper (modification in blue).
>
> In our adapted formalization, the teacher has partial observability over the environment’s state which is in our case the learner’s mental state. This strategic integration aims to clarify and strengthen the conceptual underpinning of the teaching task, facilitating a more comprehensive understanding of the teacher-learner dynamics within our proposed framework, see Section 3.3.2 (modification in blue).
>
> **The paper's objective is unclear. The authors claim, "The goal of this work is to study whether learner-specific teachers who model the learner’s internal state are more efficient than learner-agnostic ones." Given the vague problem definition, it's unclear if this captures the paper's essence. Is the primary takeaway that Bayesian ToM-teachers excel at understanding human internal state changes?**
>
> Response:
>
> We acknowledge the reviewer's concern regarding the paper's unclear formulation and objectives. In response, we have clarified these aspects in the revised introduction (modification in red).
>
> One of the main focuses of our work was to explore the utility of teachers utilizing a Theory of Mind (ToM) model compared to learner-agnostic baselines. While the superiority of ToM-teachers over learner-agnostic teaching strategies was expected, we were intrigued by the discoveries surrounding partial observation of the learner’s trajectory and the utilization of an approximate model of the learner’s policy. In fact, Section 5.2 results demonstrate that in scenarios with limited data where the teacher only has an approximate understanding of the learner's policy, employing a ToM model can be misleading. In this case, a learner-agnostic teaching strategy which aims to maximize the mean utility across all learners proves to be more useful.
>
> We chose a Bayesian model because Bayesian inference is commonly used to model theory of mind mechanisms, as extensively discussed in our related work on cognitive science. However, since our work is confined to simulated environments and virtual agents, further investigation, particularly involving human users, would be necessary to extrapolate our findings and draw conclusions about humans. Yet, we consider such research beyond the scope of this paper.

---

> > ### Author Response · Authors · 2023-11-19
> >
> > **The evaluation appears lacking. The paper notes that the teacher has a general idea of the student's policy, and during testing, the student's actions are driven by a basic decision tree. This might mean there's minimal uncertainty in the learner's behavior. It would be beneficial to incorporate human-subject experiments to see real-world impact.**
> >
> > Response:
> >
> > We appreciate the reviewer’s interest in the real-world impact of our research. At this stage of our research, our work primarily focuses on modeling teaching interactions and, therefore, centers on machine-to-machine teaching scenarios.
> > We intentionally use a known policy for the learner. We decided to work with a decision tree for simplicity as learning a policy on large 2D navigation POMDP with highly sparse reward (such as requiring the learner to grab a key to open a door) is a problem in itself and is out of the scope of this work.
> >
> > Nevertheless, we believe that the use of a trained agent would not have enhanced our findings. The aligned ToM-teacher accesses the learner’s true policy, whether trained or handcrafted. The rational ToM-teachers operate under the assumption that the learner—whether following the trained policy or the decision tree—is rational (with a certain degree that can be fitted) in its behavior. One can expect that a properly trained agent will demonstrate to a certain extent rational behavior. Consequently, the uncertainty faced by the rational ToM-teacher regarding the trained learner would resemble the uncertainty it encounters with the decision tree learner. In that sense, we believe that the use of a decision tree learner does not affect the value of our contributions.
> >
> > **While we acknowledge the significance of additional studies involving human-subject experiments to bridge the gap between simulated and real-world scenarios, we consider this to be beyond the scope of our current work.**
> >
> > Response:
> >
> > Based on points from point 1 and equation 3, the teaching policy is essentially a greedy policy. Over time, this might not be the best strategy and could lead to suboptimal results. Without a well-defined teaching task, such a policy is not well justified.
> > We concur again with the reviewer. In the fresh perspective of framing the teaching task as maximizing a reward function within a contextual MAB, the ToM-teacher's strategy aligns with choosing the greedy action (i.e., demonstration) based on an estimation of the MAB reward function (modification in blue).
> >
> > We are grateful for the references highlighted by the reviewer, which we found highly interesting and pertinent to our work. We have incorporated these papers into our related work section (modifications in green).
> >
> > [1] Srivastava et al., Assistive Teaching of Motor Control Tasks to Humans, 2022.
> > https://proceedings.neurips.cc/paper_files/paper/2022/file/b6fa3ed9624c184bd73e435123bd576a-Paper-Conference.pdf
> >
> > [2] Yu et al., COACH: Cooperative Robot Teaching, 2022.
> > https://proceedings.mlr.press/v205/yu23b/yu23b.pdf
> >
> > **How exactly do the demonstrations affect the learner('s belief, as far as I understand?)?**
> >
> > Response:
> >
> > The demonstration indeed affects the learner’s belief. The learner is a Bayesian agent which updates its belief about the state of the environment from observations, see Equation (1). When following the teacher’s demonstration, which is a sequence of actions, the learner accumulates observations of the environment. The learner’s belief is therefore updated based on these observations. This mechanism is detailed in the first paragraph of the Section 3.3.1 page 4.
> >
> > **What is the observation model of the teacher exactly?**
> >
> > Response:
> >
> > The observation model of the teacher has been clarified in the new version of the paper, Section 3.3.2 page 5 (modification in blue). The teacher only observes the learner’s internal state through a trajectory of this learner in the observation environment. The teacher also knows the initial belief of the learner in every environment, which is a uniform distribution in our setup.
> >
> > **Did you consider learning lambda?**
> >
> > Response:
> >
> > In our work, we only considered learners’ policy based on the shortest path algorithm. Therefore, this behavior corresponds to the highest degree of rationality. In that sense, we did not consider learning lambda.
> > However, had the learner's policy been learned, incorporating this parameter would have become essential to approximate the learner's behavioral model accurately. We added a precision on this point in Appendix B.3 (modification in red).

---

### Official Review · Reviewer_PobE · 2023-10-30

**Soundness:** 2 fair
**Presentation:** 2 fair
**Contribution:** 2 fair
**Rating:** 5
**Confidence:** 3

**Summary:**

This paper proposes a technique for teaching learners through demonstration based on Bayesian principles.
In particular, the teacher maintains a posterior distribution over the unknown internal state of the learner and proposes demonstrations that maximizes the utility (learner performance minus cost of demonstration) under this expectation.

Concretely, we consider the task of teaching learners to navigate in a grid world.
The learner is specified by
    - an unknown observation model (we consider deterministic observation function of three different sizes)
    - a goal (I believe considered known)
    - an initial belief (unclear over what exactly but I believe just the initial state distribution)
    - a handcrafted policy (including A* search algorithm depending on knowledge available to the learner) *given the belief, observation model, and goal*

The teacher maintains a distribution over the unknowns, which I believe is just the observation model, but slightly different baselines are considered in the experiments too.
The task of the teacher is to pick a demonstration (from a set) such that it affects the initial belief of the learner in a way that the learner's reward is optimized (while keeping the demonstration short to avoid high costs).

**Strengths:**

Modeling other agents, whether biological or artificial, is an important step towards AI methods that can tackle real-world (multi-agent) scenarios and this paper should be interesting to a significant part of ICLR community.
The proposed approach, that of Bayesian inference over the internal state of others, is both flexible as well as suitable and seems a reasonable research direction.

In particular, I believe the core contribution is how to model giving demonstrations as a teacher in the Bayesian setting should/could be done.
Furthermore, the proposed protocol (what is visible to whom between teacher and learner) is rather interesting and realistic.

To summarize, I find that this paper tackles a relevant problem with reasonable assumptions in an interesting way.

**Weaknesses:**

My main concerns are regarding the presentation and impact.

Regarding presentation, I find the formalization unclear and sometimes misaligned with the literature.
For example, the environment is introduced as a POMDP but then formalized (seemingly?) as an MDP (there are no observations?).
Additionally, the Bayes-adaptive POMDP is mentioned but then it seems that the uncertainty is limited to the state (is the belief only over the state?), which is not a problem solved by the BA-POMDP at all.
Various (important) notions are not defined, including the learner's initial belief, which the teacher is trying to affect, which makes it hard to understand what exactly the demonstrations are accomplishing.
Another example is the data observed by the teacher (tau^obs) which does not seem to be defined anywhere.

I also found the story in which the method was introduced hard to follow: it is a linear specification in which each component is discussed one at a time.
However, this is difficult to understand because it is unclear which decisions are problem specific, and which are more general in nature.
For example, assuming I correctly understood that the belief of the learner is just the (initial?) state distribution, is this specific to the problem or more generally a property of the method?
A general-to-concrete story would have made the contribution much clearer, in my opinion.

Lastly, as far as I understand, (Bayesian) ToM for teaching learners is not quite novel enough to warrant a publication on its own.
For example [1,2] have done similar things, although there exact setup (interaction between learners and teachers) differ.
Hence, I believe the demonstration is a key contribution here, though the lack of clarity of the proposed method makes it hard to accept the paper in its current version.

[1] Celikok, M. M., Murena, P. A., & Kaski, S. (2020). Teaching to learn: sequential teaching of agents with inner states. arXiv preprint arXiv:2009.06227.

[2] Peltola, T., Çelikok, M. M., Daee, P., & Kaski, S. (2019). Machine teaching of active sequential learners. Advances in neural information processing systems, 32.

**Questions:**

- How exactly do the demonstrations affect the learner('s belief, as far as I understand?)?
- What is the observation model of the teacher exactly?
- Did you consider learning lambda?

---

> ### Author Response · Authors · 2023-11-19
>
> We thank the reviewer for their constructive comments on our work. We are glad that the reviewer found that we are tackling a relevant problem and that they found our approach suitable.
>
> **My main concerns are regarding the presentation and impact.**
>
> **Regarding presentation, I find the formalization unclear and sometimes misaligned with the literature. For example, the environment is introduced as a POMDP but then formalized (seemingly?) as an MDP (there are no observations?).**
>
> Response:
>
> The learners’ environment is framed as a POMDP, yet unlike the conventional formalism, the observation functions are considered internal to the learners rather than to the POMDP. As a result, within the same environment, two learners with varying observation functions will receive different observations from the same environment states. We added in the main paper a reference to a similar work using the same formalization (modification in green).
>
> In the revised version of the paper, we've introduced a teaching contextual Multi-Arms Bandit (which is a simpler case of POMDP) that delineates the teacher’s task from that of the learners’ in Section 3.3.2. We hope that the reviewer will find this revision clarifies our method (modifications in blue).
>
> **Additionally, the Bayes-adaptive POMDP is mentioned but then it seems that the uncertainty is limited to the state (is the belief only over the state?), which is not a problem solved by the BA-POMDP at all.**
>
> Response:
>
> We referred to the BA-POMDP framework to motivate the use of a belief over the state of the environment which was introduced as a ‘belief state’ in the paper [1] “The belief state specifies the probability of being in each state given the history of action and observation experienced so far, starting from an initial belief b0.” However, as the learner uncertainty relies solely on the state of the environment, it is true that referring to Bayesian approaches of POMDP is more accurate. We thank the reviewer for this useful remark and removed the BA-POMDP reference from the paper, page 4 (modifications in red).
>
> [1] Stéphane Ross, Brahim Chaib-draa, and Joelle Pineau. Bayes-adaptive POMDPs, NeurIPS, 2007.
>
> **Various (important) notions are not defined, including the learner's initial belief, which the teacher is trying to affect, which makes it hard to understand what exactly the demonstrations are accomplishing.**
>
> Response:
>
> The initial belief of the learners is said to be uniform (Section 3.2, page 4, “Unless mentioned otherwise, we assume that the learner’s initial belief bi,j_0 on the state of Mj is uniform over the set of possible states Sj_B .“).
>
> In the revised version of the paper, the initial uniform belief of the learner, which has not yet interacted with the teacher, is considered to be an observation of the learner’s mental state (which is the hidden state of the teaching contextual MAB). This is defined in Section 3.3.2 page 5 (modification in blue) and we hope that the reviewer finds this formalization clarifies this point.
>
> In the demonstration environment, the teacher offers guidance to familiarize the learner with the new environment. Starting from an initial uniform belief, the  learner follows the teacher's demonstration –which is a sequence of actions– and updates its belief about the state of the new environment from the observations it has accumulated by following the teacher’s demonstration. Subsequently, during the evaluation phase, the learner behaves on its own but starts with the initial belief updated by the teacher’s demonstration rather than an uniform initial belief. This is detailed in the first two paragraphs of Section 3.3.1 pages 4 and 5.
>
> **Another example is the data observed by the teacher (tau^obs) which does not seem to be defined anywhere.**
>
> Response:
>
> The data observed by the teacher was defined in Section 3.3.2 paragraph 4, page 5, “From a past trajectory τ^obs = {(s_k, a^{obs}_k )}^{K−1} _{k=0} of an unknown learner L on the first environment Mobs [...]”.
>
> We thank the reviewer for highlighting this unclear point. In the new version, the teacher’s observation of the learner is more clearly defined as an observation function of the learner’s internal state, see the new Section 3.3.2 page 5 (modifications in blue).

---

> > ### Author Response · Authors · 2023-11-19
> >
> > **I also found the story in which the method was introduced hard to follow: it is a linear specification in which each component is discussed one at a time. However, this is difficult to understand because it is unclear which decisions are problem specific, and which are more general in nature. For example, assuming I correctly understood that the belief of the learner is just the (initial?) state distribution, is this specific to the problem or more generally a property of the method? A general-to-concrete story would have made the contribution much clearer, in my opinion.**
> >
> > Response:
> >
> > We appreciate the reviewer’s feedback regarding the narrative structure of our method's introduction. We understand the importance of a clear and coherent story to elucidate how each component contributes to both the general method and its problem-specific applications, in our case 2D navigation task.
> >
> > In the paper, we are already presenting a model of learners with internal states aiming at maximizing a reward function in a GC-POMDP framework. Subsequently, we detail the particularity of such learners in a context of 2D navigation. Now, in the revised version of our work, we follow the same general-to-concrete story for the teacher’s presentation. We added a clearer formulation of the teacher’s task in order to dissociate more clearly the general approach of our method from its application to 2D navigation, see Section 3.3.2 page 5 (modifications in blue).
> >
> > **Lastly, as far as I understand, (Bayesian) ToM for teaching learners is not quite novel enough to warrant a publication on its own. For example [1,2] have done similar things, although their exact setup (interaction between learners and teachers) differ. Hence, I believe the demonstration is a key contribution here, though the lack of clarity of the proposed method makes it hard to accept the paper in its current version.**
> >
> > Response:
> >
> > Both references pointed out by the reviewer are relevant to our work and deal with similar problems.
> > However, while [1] introduces learners with mental states, the objective of this work differs from ours as it addresses meta-learning issues. In this study, the teacher’s demonstrations aim to optimize the learner’s mental state to enhance its learning abilities. Conversely, in our work, the learner's mental state remains fixed and the teacher's demonstrations are aimed at assisting the learner in maximizing rewards for specific tasks.
> >
> > The setting in [2] shares similarity in the issues it addresses. In that paper, the teacher’s demonstrations aim to assist the learner in maximizing rewards for specific tasks. Additionally, they explore the utilization of a ToM model to enhance teaching. However, this study focuses on the learner’s ToM model of the teacher, used to modulate the processing of the provided demonstrations whereas we implemented a teacher’s ToM model of the learner used to select the demonstration. Furthermore, while our work concentrates on empirical results in a complex 2D navigation environment, [2] emphasizes theoretical findings in relatively simple theoretical environments.
> >
> > Finally, we consider the primary contribution of our work to be the exploration of ToM models when dealing with limited observation of learners, rather than solely the introduction of a demonstration. We found particularly interesting the discovery in Section 5.2 that, with limited observation of the learner, a teacher using an approximation of the learner’s policy is less useful than a teacher aiming to maximize the mean utility across all potential learners. This insight suggests that ToM models should be employed only when enough observation of the learners and/or precise knowledge of their policy are available. We precise this point in the introduction and in the formulation of the paper’s goals, pages 1 and 2 (modification in red).
> >
> > We appreciate the reviewer for highlighting these papers; they indeed provide valuable context for our work. We included reference [2] in our related work section (modification in green), as it closely aligns with our research. However, we determined that reference [1] was further from the scope of our work as it was more related to meta-learning.
> >
> > [1] Celikok, M. M., Murena, P. A., & Kaski, S. (2020). Teaching to learn: sequential teaching of agents with inner states. arXiv preprint arXiv:2009.06227.
> > https://arxiv.org/pdf/2009.06227.pdf
> >
> > [2] Peltola, T., Çelikok, M. M., Daee, P., & Kaski, S. (2019). Machine teaching of active sequential learners. Advances in neural information processing systems, 32.
> > https://proceedings.neurips.cc/paper_files/paper/2019/file/b98a3773ecf715751d3cf0fb6dcba424-Paper.pdf

---

> > > ### Comment · Reviewer_PobE · 2023-12-04
> > >
> > > Thank you for your response.
> > >
> > > While it helps answer some of the specific questions I had, overall it does not sufficiently address the concerns regarding clarity/significance of contribution to change my evaluation

---

### Official Review · Reviewer_sVdN · 2023-10-31

**Soundness:** 3 good
**Presentation:** 3 good
**Contribution:** 2 fair
**Rating:** 5
**Confidence:** 4

**Summary:**

Based on the theory of mind (ToM), the teacher agent builds a model of the student's mental modal and then selects the best demonstration from a pool of demonstrations that maximizes the students reward minus the cost of demonstration. Authors compared two ToM based teachers against 4 baselines teachers and 1 that represents the upper bound for performance. Empirical results shows the goal and receptive field of the student can help achieving higher returns compared to 4 baselines.

**Strengths:**

+ The paper is very well-written and easy to follow.
+ Experimental results are encouraging as teachers taking advantage of ToM to customize the guidance out-performed the ones that do not utilize ToM.
+ The approach is simple and authors provided a git repository including notebooks for fast adoption.

**Weaknesses:**

- Generalizability: The proposed ToM technique assumes access to an approximate policy of the student. Also all environments discussed in the paper were deterministic. In practice, while the set of student goals are limited, the policy they follow may be far from ideal and the presence of stochasticity may confuse the teacher further to reach a reasonable belief. Would be great to discuss these limitation in the paper.
- Computational Complexity Analysis: Given the calculation of the belief over goal x observation function, the proposed method does not scale well for more realistic scenarios. While authors left this to future works, the paper can benefit from a complexity analysis.
- Limited novelty: the main idea of the paper is not novel and similar ideas have been explored to infer agent policies and goals as cited by the authors. The main difference is to expand the inference space to include observation model.

Minor comments:
- Spell out ISL: Implicit statistical learning
- Typo: A set of states S^i => S^j

**Questions:**

- "receptive field", does the agent see behind walls? I believe the answer is yes, but would be great to clarify in the main doc.
- Why demonstrations are action sequence only? Why exclude observations? Is it due to deterministic assumption?

---

> ### Author Response · Authors · 2023-11-19
>
> We thank the reviewer for their positive evaluation of our work and for helpful comments, including spotting typos. In particular, we are glad that they found the paper easy to read and that the reviewer appreciated the notebooks.
>
> **Generalizability: The proposed ToM technique assumes access to an approximate policy of the student. Also all environments discussed in the paper were deterministic. In practice, while the set of student goals are limited, the policy they follow may be far from ideal and the presence of stochasticity may confuse the teacher further to reach a reasonable belief. Would be great to discuss these limitations in the paper.**
>
> Response:
>
> The reviewer brings up an interesting point. We introduce the rational ToM-teacher, characterized by a stochastic approximate policy of the learner. This policy is further defined by a Boltzmann temperature parameter, lambda, representing the assumed degree of rationality of the learner. A higher temperature parameter implies a noisier rationality in the learner's behavior. Therefore, in theory, with adapted temperature parameters, the rational ToM-teacher can be effective for noisily rational learners
>
> Although our experiments only involve learners following a deterministic shortest path policy, representing perfect rationality with the lowest temperature parameter, we could have employed stochastic, noisily rational learners. In such cases, learning the temperature parameter would have been necessary. We added a precision on that point in Appendix B.3 (modification in red). However, learning a good policy within a complex POMDP, such as large Minigrid environments with sparse rewards (for instance, requiring the key of a specific color to open a door), constitutes a distinct research challenge beyond the scope of this paper.
>
> **Limited novelty: the main idea of the paper is not novel and similar ideas have been explored to infer agent policies and goals as cited by the authors. The main difference is to expand the inference space to include an observation model.**
>
>
> Response:
>
> We acknowledge the reviewer's concern regarding the novelty of our work. We recognize that the introduction of our objectives may have been misleading. In the revised paper, we have clarified our focus, emphasizing the results obtained in the regime of limited observation of the learner’s behavior and ToM-teachers with an approximate understanding of the learner's policy (modification in red). Additionally, we have incorporated references highlighted by the reviewers to better underscore the novelty of our work (modifications in green). We trust that these modifications will demonstrate the significance of our findings and the novelty of our approach to the reviewer.
>
> **"receptive field", does the agent see behind walls? I believe the answer is yes, but would be great to clarify in the main doc.**
>
>
> Response:
>
> If the learner has a limited observation function, it does not see behind walls (we set the argument ‘see_through_walls=False’ to generate our environments with the MiniGrid library). We added a clarification of this point in the main document in Section 4, point on the learner (page 6) (modifications in red).
>
> **Why are demonstrations action sequences only? Why exclude observations? Is it due to a deterministic assumption?**
>
>
> Response:
>
> The observation function differs between the teacher and the learners. The teacher has full observability while the learners can have limited and different observation functions. The idea is that the same demonstration as a sequence of actions will result in different sequences of observations for two learners with different observation functions (i.e. receptive field sizes). Hence while being exposed to the same demonstration, two learners with different observation functions will acquire different knowledge of the environment.
> The demonstration, presented as a sequence of actions, serves as a method for the teacher to guide the learner in the environment. However, the teacher must infer from prior observation what the learner will perceive while being guided. We added a clarification in Section 3.3.1 page 4 (modifications in red).

---

> > ### Comment · Reviewer_sVdN · 2023-12-05
> >
> > I appreciate authors response. After reading all reviews and responses, I agree with other reviewers that the paper needs to better situate the work compared to previous efforts and highlight the key filled gaps, hence lowering my score from 6 to 5. Having said that, I encourage authors to pursue this line of work and resubmit the paper.

---

### Official Review · Reviewer_C7a4 · 2023-10-31

**Soundness:** 2 fair
**Presentation:** 2 fair
**Contribution:** 1 poor
**Rating:** 1
**Confidence:** 5

**Summary:**

The paper presents adaptive teaching strategies using Bayesian theory of mind.  Inspired by research in cognitive science, the paper proposes a goal-conditioned POMDP framework in which teachers choose demonstrations for learners under uncertainty about their beliefs. The paper presents an extended description of the model itself followed by experimental demonstrations in a simple gridworld environment.

**Strengths:**

- The paper tackles an interesting problem: how to formalize teaching goal-directed agents.
- To the extent the teaching one intends is of humans, the interest in cognitive science is laudable.

**Weaknesses:**

The paper does not engage with, or obviously contribute to the large literature on models of teaching in machine learning (or cognitive science). Specifically, the introduction is entirely focused on one qualitative theoretical perspective in the cognitive science literature on teaching, without mentioning the extensive literature on formalizing teaching and cooperation. This literature has grown large and quite mature in recent years, including extensive mathematical and computational theories from a variety of perspectives. Indeed, several papers touch on topics that are quite close to the ideas here including sequential teaching under perturbations on belief and policy, teaching as a POMDP, proofs of robustness of standard ToM reasoning in cooperative settings, etc. Indeed, I would argue that the results are not surprising given what we already know. The paper does not make contact with these results, indeed doesn't even cite many of the relevant papers.

Detailed comments:
- The introduction is poorly structured to help readers understand the literature. There has been extensive work before and after the Gweon et al paper that explores this concept.
- The introduction is almost exclusively focused on empirical research. However, there has been extensive work developing models. What is the contribution vis a vis that literature?
- "we explore the limitations of ToM models not being able to recover the learner actual internal state from its behaviour" awkward sentence.
- The contribution is rather modest. There are models for learning from observation. There are models of teaching. It appears the argument for novelty here is to do both? As noted in the literature review, this is not particularly novel either?
- The related work section is far too broad. Theory of Mind and Bayesian inference are topics that couldn't possibly fit in a related work section.
- There are a large number of related papers that are omitted. In recent years, there are several related NeurIPS papers, search for teacher or teaching or cooperation. There are older papers formalizing teaching as a POMDP problem. There are also newer papers on inferring beliefs of agents.
- It is notable that neither the introduction nor the related work discuss POMDPs.
- "we introduce a teacher equipped with a Theory of Mind (ToM) model that we refer to as ToM-teacher" Important to acknowledge most papers have a ToM component.
- "We assume that the teacher has knowledge of the learner’s uniform initial belief and has access to a behavioural model of the learner – that is an approximation of its policy πˆ – along with sets of possible goals GB and observation functions VB . These spaces are assumed discrete." These are strong assumptions that are very close to existing prior work. There are theoretical results that suggest why these assumptions are strong enough to work.
- Given the introduction focusing on cognitive science, it is surprising to see decision trees and A* as part of the learner. Is there reason to believe that these are reasonable models of humans?

I would strongly recommend that the authors review the last few years of NeurIPS (also ICML) papers for topics such as "teacher", "teaching", "cooperation" and "cooperative". Not all papers with those words will be related, but one will find quite a lot. Specifically, there are relatively theoretical papers that outline a mathematical framework and imply the results here that appeared in NeurIPS and ICML. There are also papers in NeurIPS that tested human experiments, which necessarily have errors in beliefs. Similarly, please read the literatures related to RSA (the Goodman and Frank paper cited) which have a number of interesting models and results. Please also search broadly for teaching and POMDPs as there is related work in that direction also. The current work will benefit from reconceptualization in light of these works.

**Questions:**

Please see the limitations. The big question being: what is the contribution of the current work based on prior results.

---

> ### Author Response · Authors · 2023-11-13
> **Request for concrete references**
>
> Before preparing our response to all reviewers, we would be glad if the reviewer could help us better understand the weaknesses of our work by pointing to some concrete instances of the relevant literature that we missed and that the reviewer seems to have in mind. The reviewer says that we do not mention “the extensive literature on formalizing teaching and cooperation”, they say that “This literature has grown large and quite mature in recent years, including extensive mathematical and computational theories from a variety of perspectives.” and even say that “several papers touch on topics that are quite close to the ideas here including sequential teaching under perturbations on belief and policy, teaching as a POMDP, proofs of robustness of standard ToM reasoning in cooperative settings, etc.”, but doesn’t give even a single instance of such papers that we missed.
>
> Further in the review, the reviewer says “In recent years, there have been several related NeurIPS papers, searching for ‘teacher’ or ‘teaching’ or ‘cooperation’. There are older papers formalizing teaching as a POMDP problem. There are also newer papers on inferring beliefs of agents.”. Again, mentioning a few instances of such papers that the reviewer may have in mind would help a lot.
> Finally, the reviewer says “I would strongly recommend that the authors review the last few years of NeurIPS (also ICML) papers for topics such as ‘teacher’, ‘teaching’, ‘cooperation’ and ‘cooperative’. Not all papers with those words will be related, but one will find quite a lot. Specifically, there are relatively theoretical papers that outline a mathematical framework and imply the results here that appeared in NeurIPS and ICML. There are also papers in NeurIPS that tested human experiments, which necessarily have errors in beliefs. Similarly, please read the literature related to RSA (the Goodman and Frank paper cited) which have a number of interesting models and results. Please also search broadly for teaching and POMDPs as there is related work in that direction also. The current work will benefit from reconceptualization in light of these works.”
>
> Again, the reviewer seems to have some concrete papers in mind, but does not mention them. So we would be extremely grateful to the reviewer if they could provide at least 2 or 3 of the most relevant papers they have in mind, so that we can enrich our perspective with these works if they are actually relevant, or eventually better delineate our scope if we think they are not.
>
> In particular, as the reviewer mentions "cooperation" and "cooperative", we would like to clarify that our work is not much related to the abundant literature about cooperation in multiagent systems, where indeed one can find POMDP-based formalizations and Bayesian inference mechanisms, but teacher-learner interactions are not considered. Rather, our work is inspired by cognitive sciences, and more specifically by recent works on inferential social learning (ISL) (Gweon et al., 2021), which argues that humans learn from evidence generated by others and generate useful evidence to help others learn.

---

> > ### Comment · Reviewer_3FPZ · 2023-11-14
> >
> > I'm not sure which papers Reviewer C7a4 is referring to. However, in terms of modeling teaching as a POMDP, particularly with respect to teacher-learner interactions, there is one relevant paper that was missed:
> >
> > Rafferty et al., "Faster Teaching via POMDP Planning," 2015.

---

> > ### Comment · Reviewer_C7a4 · 2023-11-17
> > **References**
> >
> > Here are a few related papers. I understand that the paper is inspired by the cognitive science literature, which is why it is surprising the authors only cite a single paper. The ideas in the Gweon paper have been developed in a long series of papers including (but not limited to): Bonawitz et al, 2022; Shafto, Goodman, & Frank, 2012; Jara-Ettinger et al., 2016. There are also works on models of trust ("epistemic trust") in the cognitive science literature. If you intend to build on cogsci, it would be ideal to demonstrate command of that literature.
> >
> > In terms of teaching and cooperation. there are again many related papers. For example, there mathematical formalizations of theory of mind reasoning underlying teaching and communication (Wang et al., 2020; Wang, Wang, & Shafto, 2020) which have details relevant to differences beliefs and neural approaches (Yuan et al., 2021 and earlier papers from that group as well as citations therein). Liu et al. (2018) consider cases in which the learner's model is not fully observable.
> >
> > With respect to teaching and POMDPs, as reviewer 3FPZ notes there is prior work by Rafferty et al (2015), with antecedent work in 2011.
> >
> > Many elements of what is proposed in the current work have been proposed previously. Reviewing the literature carefully is necessary to clarify the novel contributions, as is comparison with previous models.

---

> > > ### Author Response · Authors · 2023-11-19
> > >
> > > We appreciate the reviewer's effort in providing feedback on our work. We are very thankful for all the references provided by the reviewer which are very relevant to our current research and which greatly improved our understanding of the current literature on this subject.
> > >
> > > **The introduction is poorly structured to help readers understand the literature. There has been extensive work before and after the Gweon et al paper that explores this concept. The introduction is almost exclusively focused on empirical research. However, there has been extensive work developing models. What is the contribution vis a vis that literature?**
> > >
> > > Response:
> > >
> > > Our main focus was indeed to leverage the fine-grained understanding of cognitive science research on the human mechanisms, such as ToM, used to transfer information between human individuals. We acknowledge that our work relied on existing modeling of ToM mechanisms, on one hand, and teaching, on the other hand, and we referred to this existing literature in our related work and added the references highlighted by the reviewers (modification in green).
> > >
> > > However, our aim was to provide empirical evidence that demonstrates how employing a ToM model improves the effectiveness of a machine teacher for a machine learner. This differs from existing work, which primarily concentrates on using a ToM model to enhance the learners themselves. Furthermore, our aim was to offer empirical results regarding the impact of an inaccurate prior held by the teacher on the learner's policy, on the ToM teacher’s utility. We clarified this point in the related work and introduction in which we added some of the references pointed by the reviewer (modification in green).
> > >
> > > **"we explore the limitations of ToM models not being able to recover the learner's actual internal state from its behaviour" awkward sentence.**
> > >
> > > Response:
> > >
> > > We appreciate the reviewer's feedback on the stylistic aspect and have updated the sentence accordingly in the new version (modifications in red).
> > >
> > > **The contribution is rather modest. There are models for learning from observation. There are models of teaching. It appears the argument for novelty here is to do both? As noted in the literature review, this is not particularly novel either?**
> > >
> > > Response:
> > >
> > > We acknowledge the reviewer’s concern regarding the perceived modesty of our results. We are grateful to the reviewer for sharing relevant work on modeling ToM and how it can be leveraged in order to improve teacher-learner interactions. In fact, we added some of these references to our introduction and related work (modification in green) as the Bayesian models used for modeling ToM were similar to ours.
> > >
> > > However, in the references pointed out by the reviewer, mainly Shafto et al. 2012 and Bass et al. 2022, the main focus of the works is to model ToM in the learner in order to infer the goal, intention and desire of the teacher. This way, the learner is able to identify knowledgeable and well-intended teachers. Conservely, in our work, the learner does not process a ToM model of the teacher. It is the teacher who processes a ToM of the learner. Hence, ToM is not used to learn but rather to help learning. Our paper presents better teachers with ToM whereas the works highlighted by the reviewer present better learners with ToM.
> > > The related work section is far too broad. Theory of Mind and Bayesian inference are topics that couldn't possibly fit in a related work section.
> > >
> > > **It is notable that neither the introduction nor the related work discuss POMDPs.**
> > >
> > > Response:
> > >
> > > We opted to concentrate on the cognitive science literature, particularly exploring how theory of mind has been examined and modeled, with a specific emphasis on Bayesian inference models.
> > >
> > > However, we acknowledge the relevance of the literature on POMDPs suggested by other reviewers, and as such, we have incorporated the suggested works into the revised version of the paper (modification in green).

---

> ### Author Response · Authors · 2023-11-19
>
> **"we introduce a teacher equipped with a Theory of Mind (ToM) model that we refer to as ToM-teacher" Important to acknowledge most papers have a ToM component.**
>
> Response:
>
> Our objective was to explicitly model the ToM employed by the teacher to instruct the learner. While there are numerous models demonstrating ToM capacities and incorporating ToM components,  to our knowledge, in most of the literature, the ToM model is defined from the learner toward the teacher and is used to modulate how the learner processes the information provided by the teacher. In fact, none of them explicitly defines a model of ToM from the teacher to the learner and shows how this model can be leveraged to modulate teaching in itself. While the current literature focuses on how ToM is used to learn, Gweon research and our work focus on how ToM is used to help others learn.
>
> **Given the introduction focusing on cognitive science, it is surprising to see decision trees and A-star as part of the learner. Is there reason to believe that these are reasonable models of humans?**
>
> Response:
>
> We do not claim that decision trees and A* algorithms are reasonable models of humans. However, we intentionally use a known policy for the learner in order to analyze the impact of inaccurate prior on the learner’s policy. We specifically use a decision tree, due to the complexity of learning in a large 2D navigation POMDP with sparse rewards—a challenge beyond this work's scope.
>
> We could have used a trained agent. No matter if a trained or handcrafted policy is used, the ToM-teacher assumes the learner behaves rationally. We anticipate a properly trained agent to exhibit rational behavior, creating similar uncertainty for the ToM-teacher, whether the learner follows a trained policy or a decision tree. Thus, we believe that employing a decision tree learner does not impact the value of our contributions.

---

> > ### Comment · Reviewer_C7a4 · 2023-11-22
> > **Response to the response**
> >
> > Thanks for your response.
> >
> > Regarding theory of mind: it is symmetric. You can either think about the learner reasoning about the teacher or the teacher reasoning about the learner. This is a point that has been made clearly in the papers that the authors don't cite.
> >
> > Similarly, the result about discrepancies in teacher's beliefs about the learner has also appeared. In fact, we know quite a bit about the consequences of discrepancies from prior simulations and theoretical results. Unfortunately, the relevant works are not cited.
> >
> > It is strange to talk so much about theories from cogsci, but not have the goal of modeling (even idealized) humans. If you have a specific hypothesis, it seems more direct to argue for that rather than partially pursuing the analogy to humans.
> > For example, "Our main focus was indeed to leverage the fine-grained understanding of cognitive science research on the human mechanisms" and "We do not claim that decision trees and A* algorithms are reasonable models of humans." are inconsistent.
> >
> > "This work is a first step towards social machines that teach us and each other," This sentence is patently false. As all the reviewers note, there are dozens of prior works that address the question of social machines that teach.
> >
> > I stand by my assessment that there needs to be a full reconceptualization of the work here.

---

### Official Review · Reviewer_dJ9E · 2023-11-01

**Soundness:** 4 excellent
**Presentation:** 4 excellent
**Contribution:** 2 fair
**Rating:** 3
**Confidence:** 2

**Summary:**

This paper studied machine teaching of reinforcement learners in the partially observable MDP (POMDP) setting. The teacher models the student learner's Theory of Mind (ToM) with Bayesian framework. In particular, there is a simple POMDP environment where teacher interacts with the student to understand the mental state of the student, and then applies the learned knowledge to teach the student in a more complex POMDP environment. The paper showed that when teacher's prior is aligned with the student's mental state, then the teacher demonstration can better help the student learn a good policy. Extensive empirical results validated the claims made in the paper.

**Strengths:**

(1) The paper proposed to use the Bayesian framework to model student's mental state, and then utilizes the learned knowledge to help improve the learning process of the student in a complicated POMDP environment. This approach is reasonable and sound.

(2) The paper performed extensive empirical explorations to demonstrate that the proposed method indeed helped speed up the learning process of the student model. The results look great and are convincing.

**Weaknesses:**

(1) It's not clear why the method requires two learning environment - a simple one for teacher to interact with the learner and gain knowledge of the student's metal state; and another more complex environment where the teacher performs real teaching. Is it possible to unify these two and let the teacher teach on the fly as it interacts with the student?

(2) If there are significant distinctions between simple and complex environment. How does it affect the teaching process? Is the knowledge learned by the teacher in the simple environment going to become less effective?

(3) The teacher requires full knowledge of the underlying POMDP. This is really unrealistic unless in very specific domains. This requirement limits the applicability of the proposed method.

(4) The idea of the paper does not seem novel to me, and the results are not surprising at all. Of course, if the teacher learned prior for student's metal state aligns with the real mental state, then it's going to teach better. It would be much more interesting if the authors can provide theoretical justification of the proposed method through some Bayesian inference theory.

**Questions:**

Please see weaknesses above.

---

> ### Author Response · Authors · 2023-11-19
>
> We thank the reviewer for their time and effort in evaluating our work, and are glad that the reviewer found the soundness and presentation of our paper excellent.
>
> **1) It's not clear why the method requires two learning environments - a simple one for the teacher to interact with the learner and gain knowledge of the student's mental state; and another more complex environment where the teacher performs real teaching. Is it possible to unify these two and let the teacher teach on the fly as it interacts with the student?**
>
> Response:
>
> Teaching 'on the fly' while interacting with the learner is possible but still requires sequences of observation, demonstration selection, and evaluation. We decided to split the framework into two environments to separate the Theory of Mind (ToM) modeling (teacher modeling the learner’s internal state) from the teaching (teacher leveraging its model of ToM to provide a demonstration).
>
> **2) If there are significant distinctions between simple and complex environments. How does it affect the teaching process? Is the knowledge learned by the teacher in the simple environment going to become less effective?**
>
> Response:
>
> The first environment’s complexity can vary, and in additional experiments which we included in Appendix G of the revised version, we show that actually performing the observation stage on a simpler environment causes more ambiguity in the learner’s behavior, causing the teacher to have more difficulty to identify the learner’s goals and beliefs. Indeed, since the learner can have different widths of sensory capacity, being observed in a larger, more complex environment can actually help better identify its sensors.
> However, if the observation environment was overly complex for the learner to engage with objects, its behavior would not be indicative enough of its goal, making inferring internal states impossible. Hence, the need for a reasonably simple environment in which all the learners can interact with the objects without the help of the teacher.
>
> On the other hand, the second environment, where the teacher conducts teaching, needs to be complex enough for some learners to require the teacher’s assistance to achieve the task. If learners can easily accomplish tasks without the teacher's help, providing an empty demonstration would be an optimal solution: no teaching would be needed. Hence, the need for a ‘complex’ environment. Finally, note that a learner’s internal state (goal and observation function) remains consistent across different environments and is independent from the complexity of those environments. Consequently, the insights gained by the teacher during the observation phase can be leveraged for teaching in any environment, regardless of its complexity.
> We added in Appendix G of the revised version of the paper a precision on this point discussing the influence of the observation environment size (modifications in orange).
>
> **3) The teacher requires full knowledge of the underlying POMDP. This is really unrealistic unless in very specific domains. This requirement limits the applicability of the proposed method.**
>
> Response:
>
> Indeed, we assume the teacher's access to the POMDP. Specifically, our method requires the teacher to predict a learner's reward in the environment based on a given policy conditioned by a particular belief. In our method, this involves simulating trajectories within the environment, which relies on the teacher's knowledge of the underlying POMDP. However, for reward prediction, the teacher can use a pre-trained model. We could consider a model trained to predict the reward of a learner following a particular belief-conditioned policy (the teacher’s approximation of the learner’s policy), therefore taking a belief as input. Yet, this would require access to a large amount of trajectories of such learners with various beliefs following the teacher’s approximate policy.
>
> We chose to focus on uncertainty on the learner’s mental state rather than on the POMDP from the teacher’s point of view. We acknowledge the interest of considering teachers with partial observability but we consider that this study is out of this work’s scope.

---

> > ### Author Response · Authors · 2023-11-19
> >
> > **4) The idea of the paper does not seem novel to me, and the results are not surprising at all. Of course, if the teacher learned prior for the student's mental state aligns with the real mental state, then it's going to teach better. It would be much more interesting if the authors can provide theoretical justification of the proposed method through some Bayesian inference theory.**
> >
> > Response:
> >
> > We understand the reviewer’s concern about the non-surprising nature of the results when interpreted this way. It is definitely obvious that an omniscient teacher will be best, and that an aligned teacher, as defined in the paper, will perform second best. Indeed, these teachers possess knowledge that would not be accessible in a realistic scenario.
> >
> > The core of our results, which in our opinion are more interesting and surprising, are related to the rational and non-adaptive teachers. These teachers are based upon more realistic assumptions about the learner, and we showcase how being rational is superior when having access to full trajectories of the learner, and inferior to non-adaptive teachers when having access to limited trajectories. It quantifies the right amount of information needed for the teacher to provide accurate adaptive teaching to the learner, which we believe is a hard problem in teacher-learner scenarios.
> >
> > We hope this clarification can help the reviewer understand the relevance of our results. We clarified this positioning in the introduction in the formulation of the paper’s objectives (modification in red).

---

### Author Response · Authors · 2023-11-19

We express gratitude to all the reviewers for their feedback. Overall, they highlighted some pertinent work that was absent in our initial paper, questioned the formalization of the environment from the teacher’s viewpoint, sought clarification on our research objectives, and raised concerns regarding the impact of the environment’s size.

To address these points from the reviewers, the key revisions we made in the paper encompass:

*  GREEN: Incorporating the relevant literature recommended by the reviewers into the related work section.
*  BLUE: Formalizing  the teacher’s task as a contextual Multi-Arms Bandit (MAB), where the state represents the learner’s mental state observed by the teacher solely through an observation function (context) which returns a trajectory of the unknown learner in the observation environment, as well as the learner’s initial belief. This formalization clarifies the utilization of a teacher’s belief regarding the learner’s internal state and distinguishes our approach from its application to 2D navigation.
*  ORANGE: Including an experiment in Appendix G that examines the accuracy of teachers’ Theory of Mind (ToM) models when observing learners in observation environments of varying sizes.
*  RED: Addressing clarity concerns raised by the reviewers on the method and more importantly on the paper’s contribution and positioning.

For ease of reference to the changes we made in the manuscript, we use the color code given by the corresponding dot points.

---

### Meta-Review · Area_Chair_TmCT · 2023-12-06

**Metareview:**

The paper proposes a Bayesian Theory of Mind framework where a teacher agent infers the learner’s internal state to provide adaptive teaching demonstrations. The reviewers acknowledged that the paper tackles an important problem of teaching goal-directed agents, and the proposed Bayesian framework to model the learner’s internal state is promising. However, the reviewers pointed out several weaknesses in the paper and shared common concerns. We want to thank the authors for their detailed responses. Based on the raised concerns and follow-up discussions, unfortunately, the final decision is a rejection. Nevertheless, the reviewers have provided detailed and constructive feedback. We hope the authors can incorporate this feedback when preparing future revisions of the paper.

**Justification For Why Not Higher Score:**

The reviewers pointed out several weaknesses in the paper. There was a consensus among the reviewers that the work is not yet ready for publication.

**Justification For Why Not Lower Score:**

N/A

---

### Decision · Program_Chairs · 2024-01-16

Reject